# The formation of nitro-aromatic compounds under high NO$_x$ and anthropogenic VOC conditions in urban Beijing, China

Yujue Wang[1], Min Hu[*,1,5], Yuchen Wang[3], Jing Zheng[1], Dongjie Shang[1], Yudong Yang[1], Ying Liu[1,5], Xiao Li[1], Rongzhi Tang[1], Wenfei Zhu[6], Zhuofei Du[1], Yusheng Wu[1], Song Guo[1], Zhijun Wu[1], Shengrong Lou[6], Mattias Hallquist[2], and Jian Zhen Yu[*,3,4]

[1]State Key Joint Laboratory of Environmental Simulation and Pollution Control, College of Environmental Sciences and Engineering, Peking University, Beijing 100871, China

[2]Department of Chemistry and Molecular Biology, University of Gothenburg, Gothenburg, Sweden

[3]Environmental Science Programs, Hong Kong University of Science & Technology, Hong Kong, China

[4]Department of Chemistry, Hong Kong University of Science & Technology, Hong Kong, China

[5]Beijing Innovation Center for Engineering Sciences and Advanced Technology, Peking University, Beijing 100871, China

[6]Shanghai Academy of Environmental Sciences, Shanghai 200233, China

*Correspondence to*: Min Hu (minhu@pku.edu.cn); Jian Zhen Yu (jian.yu@ust.hk)

**Abstract.** Nitro-aromatic compounds (NACs), as important contributors to the light absorption by brown carbon, have been widely observed in various ambient atmospheres; however, their formation in urban atmosphere was little studied. In this work, we report an intensive field study of NACs in summer 2016 at an urban Beijing site, characterized by both high-NO$_x$ and anthropogenic VOCs dominated conditions. We investigated the factors that influence NAC formation (e.g., NO$_2$, VOC precursors, RH and photolysis) through quantification of 8 NACs, along with major components in fine particulate matter, selected volatile organic compounds and gases. The average total concentration of the quantified NACs was 6.63 ng/m$^3$, higher than those reported in other summertime studies (0.14- 6.44 ng/m$^3$). 4-Nitrophenol (4NP, 32.4%) and 4-nitrocatechol (4NC, 28.5%) were the top two most abundant NACs, followed by methyl-nitrocatechol (MNC), methyl-nitrophenol (MNP) and dimethyl-nitrophenol (DMNP). The oxidation of toluene and benzene in the presence of NO$_x$ was found to be a more dominant source of NACs than primary biomass burning emissions. The NO$_2$ concentration level was found to be an important factor influencing the secondary formation of NACs. A transition from low- to high-NO$_x$ regimes coincided with a shift from organic- to inorganic-dominated oxidation products. The transition thresholds were NO$_2$~20 ppb for daytime and NO$_2$~25 ppb for nighttime conditions. Under low-NO$_x$ conditions, NACs increased with NO$_2$, while the NO$_3^-$ concentrations and (NO$_3^-$)/NACs ratios were lower, implying organic-dominated products. Under high-NO$_x$ conditions, NAC concentrations did not further increase with NO$_2$, while the NO$_3^-$ concentrations and (NO$_3^-$)/NACs ratios showed increasing trends, signaling a shift from organic- to inorganic-dominated products. Nighttime enhancements were observed for 3M4NC and 4M5NC while daytime enhancements were noted for 4NP, 2M4NP and DMNP, indicating different formation pathways for these two groups of NACs. Our analysis suggested that the aqueous-phase oxidation was likely the major formation pathway of 4M5NC and 3M5NC while photo-oxidation of toluene and benzene in the presence of NO$_2$ could be more important for

the formation of nitrophenol and its derivatives. Using the (3M4NC+ 4M5NC)/4NP ratios as an indicator of the relative
contribution of aqueous-phase and gas-phase oxidation pathways to NAC formation, we observed that the relative
contribution of aqueous-phase pathways increased at elevated ambient RH and remained constant at RH> 30%. We also
found that the concentrations of VOC precursors (e.g., toluene and benzene) and aerosol surface area acted as important
factors in promoting NAC formation, and photolysis as an important loss pathway for nitrophenols.

**1    Introduction**
Organic nitrogen, including nitro-aromatic compounds (NACs), N-heterocyclic compounds, amines and other organic
nitrate compounds containing ($-NO_2$) or ($-NO_3$) functional groups, represent an important fraction of ambient organic
aerosols (Laskin et al., 2009; Wang et al., 2017b; Chow et al., 2016; Ge et al., 2011; Ng et al., 2017). Among organic
nitrogen, NACs, with the -$NO_2$ and -OH functional groups attached to an aromatic ring, have gained much attention due to
their light absorbing property and impacts on human health (Mohr et al., 2013; Lin et al., 2017). NACs, including
nitrophenols (NPs), nitrocatechols (NCs) and their derivatives, are important contributors to the light absorption by brown
carbon (BrC) (Mohr et al., 2013; Teich et al., 2017; Zhang et al., 2013; Xie et al., 2017), contributing 50-80% of the total
visible light absorption by BrC emitted from biomass burning (Lin et al., 2017). Moreover, NACs also lead to mutagenesis
and genotoxicity, thus posing a threat to human health (Purohit and Basu, 2000; Huang et al., 1995).
NACs have been widely observed in various ambient atmospheres, including urban, suburban, rural, as well as
background environments, with the quantified concentrations varying from 0.1 ng/m$^3$ in rural background areas to 147.4
ng/m$^3$ in urban atmospheres (Iinuma et al., 2010; Teich et al., 2017; Zhang et al., 2010; Mohr et al., 2013; Chow et al., 2016;
Wang et al., 2018b). Combustion processes, especially biomass burning, were the most important primary sources of NACs
(Harrison et al., 2005; Wang et al., 2018b). The emission factors of NACs from biomass burning were estimated 0.8-11.1
mg/kg (Wang et al., 2017a; Hoffmann et al., 2007). Field observation studies indicated NACs are usually associated with
fresh or aged biomass burning aerosols, which contributed 10- 21% of the total NACs in ambient aerosols (Chow et al., 2016;
Kitanovski et al., 2012; Mohr et al., 2013; Iinuma et al., 2010; Wang et al., 2018b). Apart from primary emissions from
biomass burning, NACs could also be formed via the oxidation of volatile organic compounds (VOCs) containing a benzene
ring (e.g., cresol, catechol, methylcatechol) released by biomass burning in smoke plumes (Iinuma et al., 2010; Claeys et al.,
2012). Methyl-nitrocatechols (MNCs) could originate from $NO_x$ oxidation of methylated cresol or methylcatechols, which
are released during biomass burning as thermal degradation products of lignin (Iinuma et al., 2010; Finewax et al., 2018;
Olariu et al., 2002). 4-Nitrocatechol could be formed via the OH-initiated oxidation of guaiacol, an abundant methoxyphenol
emitted from biomass burning, in the presence of $NO_2$ (Lauraguais et al., 2014). However, under high-$NO_x$ conditions, this
pathway seems to be of minor importance to nitrocatechol formation; instead, nitroguaiacols were formed as the major
products (Lauraguais et al., 2014).
In urban atmosphere, aromatic VOCs such as benzene, toluene, and xylenes are expected to be important precursors to
NAC formation (Harrison et al., 2005). The main reactions leading to the secondary formation of NPs, NCs,
methyl-nitrophenols (MNPs) and MNCs are shown in Figure 1 (Jenkin et al., 2003; Vione et al., 2001; Vione et al., 2004;
Vidovic et al., 2018). Nitrophenols and their derivatives (e.g., MNPs) could originate through gas-phase oxidation of phenol,
benzene and toluene by OH or $NO_3$ radicals in the presence of $NO_2$ (Harrison et al., 2005; Yuan et al., 2016; Sato et al., 2007;
Ji et al., 2017; Olariu et al., 2002). Nitrocatechols dominated the composition of NACs formed in benzene/$NO_x$ system (Xie
et al., 2017). The NC formation could be initiated by OH or $NO_3$ radicals to form $\beta$-hydroxyphenoxy/$o$-semiquinone radicals,
which then react with $NO_2$ to form the final products (Finewax et al., 2018). Compared with the gas-phase formation of
NACs, the formation pathway via aqueous-phase aromatic nitration is less well understood (Kroflic et al., 2018).
Nitrophenols could be formed through the hydroxylation and nitration of benzene in the presence of nitrite/nitrous acid or
photo-nitration of phenol upon UV irradiation of nitrite in aqueous solutions (Vione et al., 2004; Vione et al., 2001). It has
been suggested that nighttime aqueous-phase oxidation is an important formation pathway for methyl-nitrocatechols,
especially in polluted high-$NO_x$ environments and in presence of acidic particles (pH around 3) (Vidovic et al., 2018). The
proposed aqueous-phase formation processes of MNCs include electrophilic substitution route and consecutive oxidation and
conjugated addition route (Frka et al., 2016; Vidovic et al., 2018). The loss pathways for NACs are proposed to include
photolysis and reactions with OH, $NO_3$ radicals or chlorine atoms (Atkinson et al., 1992; Bejan et al., 2007; Bejan et al.,
2015; Chen et al., 2011; Yuan et al., 2016; Hems and Abbatt, 2018).
However, few observational field studies have been conducted to investigate the formation of NACs in urban
atmospheres. In this work, we report results from an intensive field campaign conducted in summertime Beijing, aiming to
gain understanding of ambient concentration variation characteristics of NACs, relative importance of various proposed
formation pathways and major influence factors in high $NO_x$ and anthropogenic VOCs dominated urban atmospheres. A
group of 8 NACs (NPs, MNPs, dimethyl-nitrophenols, DMNPs, NCs and MNCs) in 19 day samples and 19 night samples
were quantified using high performance liquid chromatography- mass spectrometry (HPLC-MS). Additional data of
inorganic aerosol constituents, VOC precursors, inorganic gases and meteorological parameters were also obtained and
analyzed to aid the investigation of the secondary formation pathways of NACs and controlling factors. This work provides
insights into the secondary formation of NACs in high $NO_x$ and anthropogenic VOCs dominated urban environments.

## 2 Methods

### 2.1 Sample collection

As part of the bilateral Sweden-China framework research program on 'Photochemical smog in China', an intensive field campaign was conducted in Beijing, aiming to improve the understanding on secondary chemistry during photochemical smog events in China (Hallquist et al., 2016). The campaign was conducted at Changping (40.14° N, 116.11° E), a regional site northeast of Beijing urban area, from May 15 to June 5, 2016. During this period, the site was influenced by anthropogenic pollutants from Beijing urban areas and under high-$NO_x$ conditions, as suggested by field measurement evidence reported in previous publications related to this campaign (Tang et al., 2018; Wang et al., 2018a). During May 17-June 5, the daily average concentrations of benzene, toluene and $NO_x$ were 66-922 ppt, 47-1344 ppt and 4.0-32.3 ppb, respectively.

Day and night ambient $PM_{2.5}$ (particles with aerodynamic diameter less than 2.5 μm) samples were collected on prebaked quartz fiber filters (Whatman Inc.) and Teflon filters (Whatman Inc.) using a high-volume sampler (TH-1000C, Tianhong, China) and a 4-channel sampler (TH-16A, Tianhong, China). The sampling flow rates were 1.05 $m^3$/min and 16.7 L/min, respectively. The daytime samples were collected from 8:30 to 17:30 LT (UTC+8) and the nighttime ones from 18:00 to 8:00 LT (UTC+8) the next morning. Field blank samples were collected by placing filters in the samplers with the pump off for 30 min.

### 2.2 Quantification of NACs

An aliquot of 25 $cm^2$ was removed from each quartz fiber filter sample and extracted in ultrasonic bath three times using 3, 2 and 1 mL methanol containing 30 μL saturated EDTA solution in methanol-acetic acid consecutively, each time for 30 min. The extracts were then filtered through a 0.25 μm polytetrafluoroethylene (PTFE) syringe filter (Pall Life Sciences), combined, and evaporated to dryness under a gentle stream of high-purity nitrogen. The dried samples were re-dissolved in 50 μL methanol/water (1:1) containing 100 ppb 4-nitrophenol-2,3,5,6-$d_4$ as internal standard. The solution was centrifuged and the supernatant was used for analysis, using Agilent 1260 LC system (Palo Alto, CA) coupled to QTRAP 4500 (AB Sciex, Toronto, Ontario, Canada) mass spectrometer. The LC-MS system was equipped with an electrospray ionization (ESI) source operated in negative mode. More details of the extraction and optimized MS parameters have been described in our previous study (Chow et al., 2016).

Chromatographic separation was performed on an Acquity UPLC HSS T3 column (2.1 mm×100 mm, 1.8 μm particle size; Waters, USA) with a guard column (HSS T3, 1.8 μm). The column temperature was kept at 45 °C and the injection volume was 5.0 μL. The mobile eluents were (A) water containing 0.1% acetic acid (v/v) and (B) methanol (v/v) containing 0.1% acetic acid at a flow rate of 0.19 mL/min. The gradient elution was set as follows: started with 1% B for 2.7 min; increased to 54% B within 12.5 min and held for 1.0 min; then increased to 90% B within 7.5 min and held for 0.2 min; and

finally decreased to 1% B within 1.8 min and held for 17.3 min until the column was equilibrated. Chromatograms of NAC
standards and an ambient sample are shown in Figure S1.

The quantified NAC species are listed in Table 1. The NACs were identified and quantified using the [M-H]$^-$ ions in the

extracted ion chromatogram (EIC), using authentic standards or surrogates with the same molecular formula (Table 1). The
standards included: 4-nitrocatechol (4NC), 4-nitrophenol (4NP), 2-methyl-4-nitrophenol (2M4NP), 3-methyl-4-nitrophenol
(3M4NP)    and    2,6-dimethyl-4-nitrophenol    (2,6DM4NP)    from    Sigma–Aldrich    (St.    Louis,    MO,    USA);
4-methyl-5-nitrocatechol (4M5NC) from Santa Crutz Biotech (Dallas, TX, USA). The recoveries of the target NACs were
91-106%. 4M5NC was employed as a surrogate standard to quantify 3M5NC and 3M6NC. However, a recent study
suggested that no 3M6NC could be detected in ambient aerosols and the MNC isomer could be an incorrect assignment of
3M4NC as 3M6NC (Frka et al., 2016). We cannot exclude the possibility of MNC isomer as 3M4NC due to a lack of
authentic standards. Employing 4M5NC as a surrogate standard, the concentrations of 3M6NC could be obviously
underestimated due to its poor ionization under ESI condition compared with that of 4M5NC (Frka et al., 2016). The
concentration of dimethyl-nitrophenol (DMNP) was the sum of three isomers. 2,6DM4NP was identified based on its
retention time matching that of the authentic standard (Figure S1), while we cannot exclude the possibility of the other two
DMNP isomers as ethylnitrophenols or methoxylated isomers.
**2.3 Other online and offline measurements**

Other online and offline instruments were also employed to obtain related database, which has been introduced in

details in our previous paper (Wang et al., 2018c). In brief, a high resolution time-of-flight aerosol mass spectrometer (AMS)
was used to measure the chemical composition of PM$_1$ (Zheng et al., 2017). The aerosol surface area was calculated based on
the measurements of particle number and size distribution by a scanning mobility particle sizer (SMPS, TSI 3936) and an
aerosol particle sizer (APS, TSI 3321) (Yue et al., 2009; Wang et al., 2018a). VOCs were measured by a
proton-transfer-reaction mass spectrometer (PTR-MS). Gaseous NH$_3$ was measured using a NH$_3$ analyzer (G2103, Picarro,
California, USA) (Huo et al., 2015). Meteorological parameters, including relative humidity (RH), temperature, wind
direction and wind speed (WS) were continuously monitored by a weather station (Met one Instrument Inc.) during the
whole campaign.

Organic carbon (OC) and element carbon (EC) were measured on the quartz fiber filter samples using a thermal/optical

carbon analyzer (Sunset Laboratory). The organic matter (OM) concentration was calculated by multiplying OC by 1.6
(Turpin and Lim, 2001). The Teflon filter samples were used to quantify the water soluble inorganic ions by an ion
chromatograph (IC, DIONEX, ICS2500/ICS2000) following procedures described in Guo et al. (2010). Aerosol acidity and
liquid water content (ALWC) was then calculated using the ISORROPIA-II thermodynamic model. ISORROPIA-II was
operated in forward mode, assuming the particles are "metastable" (Hennigan et al., 2015; Weber et al., 2016; Guo et al.,
2015). The input parameters included: ambient RH, temperature, particle phase inorganic species ($SO_4^{2-}$, $NO_3^-$, $Cl^-$, $NH_4^+$, $K^+$,
$Na^+$, $Ca^{2+}$, $Mg^{2+}$), and gaseous $NH_3$. More details and validation of the thermodynamic calculations have been described in
our previous paper (Wang et al., 2018c).

**2.4 Estimation of the gas-phase NACs**

The concentrations of gas-phase NACs were not measured in this study. They were calculated based on the measured
particle-phase NAC concentrations and equilibrium absorption partitioning theory (Pankow, 1994a, b; Pankow et al., 2001)
(Eqs. 1, 2):

$$F_p = (1 + \frac{C^*}{C_{OA}})^{-1} = \frac{c_p}{c_g + c_p} \qquad \text{(Eq. 1)}$$

where $F_p$ is the fraction of NACs in the particle-phase. $C_{OA}$ is the concentration of organic aerosols (OA), calculated as OC
multiplied by 1.6. $c_g$ and $c_p$ are the concentrations of NACs in gas phase and particle phase, respectively. $C^*$ is the effective
saturation mass concentration ($\mu g/m^3$), and is calculated using Eq. 2:

$$C^* = \frac{M 10^6 \zeta P_V}{760RT} \qquad \text{(Eq. 2)}$$

where M is the molecular weight of NACs (g/mol). $\zeta$ is the activity coefficient of the species (assumed =1). $R$ is the gas
constant (8.314 J/(mol K)), $T$ is the temperature (K), and $P_v$ (Pa) is the saturation pressure. $P_v$ at the average temperature
during the campaign (296 K) is calculated using the multiphase system online property prediction tool developed by
University of Manchester (UManSysProp, http://umansysprop.seaes.manchester.ac.uk). The vapor pressures were estimated
using Nannoolal approach (Nannoolal et al., 2008), and the boiling points were estimated using the Joback and Reid
approach (Joback and Reid, 1987).
The estimated $P_v$, $F_p$ and gas-phase concentrations of NACs are listed in Table S1. 4NP and methyl-nitrophenols
(2M4NP and 3M4NP) were predicted to be mainly in the gas phase ($F_p < 10\%$) while DMNP, 4NC and MNC (3M6NC,
3M5NC and 4M5NC) were mainly in the particle phase ($F_p > 60\%$). The gas-phase DMNP and MNC ($F_p > 95\%$) will not be
further discussed in this study. While the equilibrium model gives reasonable estimation of $F_p$ and gas-phase concentrations
for nitrocacatechols, it overestimates the vapor pressure of NPs by several orders of magnitude (Bannan et al., 2017). The
estimated $F_p$ (0.83%) was obviously lower than the measured values for 4NP. For example, Cecinato et al. (2005) measured
$F_p$ of 4NP and 3M4NP to be 82% and 78%, respectively in downtown Rome; Le Breton et al. (2018) reported $F_p$ of
nitrophenol at ~17% using Chemical-Ionization Mass Spectrometer (CIMS) coupled with the Filter Inlet for Gases and
AEROsols (FIGAERO) during this campaign. We note that CIMS could not distinguish the isomers (e.g., 2NP) of 4NP,
however, the measured $F_p$ values showed us the range of particulate fraction of 4NP during the campaign. The equilibrium
absorption partitioning model could underestimate the $F_p$ of 4NP by ~20 times. Thus, the gas-phase 4NP concentration was
roughly calculated using the measured $F_p$ (17%) by FIGAERO-CIMS (Le Breton et al., 2018).

Gas-phase NACs could also dissolve into the aqueous-phase particles. The concentrations dissolved into the aqueous

phase ($C_{aq}$) were estimated by Henry's law (Sander, 2015). Henry constants were obtained from Sander et al. (2015) and
ALWC was estimated using ISORROPIA-II (see section 2.4). The estimated $C_{aq}$ of 4NP and 3M4NP were 4.4E-4 and
2.4E-5 ng/m$^3$, contributing <0.02% to their concentrations in particle phase. The contribution of dissolution into
aqueous-phase particles for NC and MNC is expected to be lower, due to the much lower gas-phase concentrations than that
of 4NP. For this reason, we will not further consider the dissolution of NACs into particle aqueous phase.
**3       Results and Discussion**
**3.1 Concentration and composition of NACs**

The average concentration of quantified NACs was 6.63 ng/m$^3$, ranging from 1.27 to 17.70 ng/m$^3$ in summer in Beijing.

Figure 2 compares the total NAC concentrations across this and prior studies, and the individual NAC concentrations are
compared in Figure S2 and Table S2. The total NAC concentration in this work was higher than those measured in other
studies conducted in summer in mountain, rural or urban environments (Teich et al., 2017; Kitanovski et al., 2012; Kahnt et
al., 2013; Zhang et al., 2013; Chow et al., 2016; Wang et al., 2018b), and comparable to those reported in the studies in
summertime Wangdu, China (Teich et al., 2017; Wang et al., 2018b) (Figure 2). Most NAC species (NC, MNP and MNC),
except for DMNP and NP, also showed elevated concentrations in Changping, compared with those reported in other
summertime studies (Figure S2). Influenced by the outflow from urban Beijing air masses, the site was under typical
high-NO$_x$ conditions (Wang et al., 2018a), implying abundant potential secondary formation of NACs during the observation
period. A recent study suggested that nocturnal biogenic VOCs (BVOCs) oxidation would transfer from low- to high-NO$_x$
regimes and nearly all the BVOCs would be oxidized by NO$_3$ radicals, at a NO$_x$/BVOCs ratio higher than 1.4 (Edwards et al.,
2017). If we approximate the BVOC concentrations to be the sum of isoprene, MVK+MACR (methyl vinyl ketone and
methacrolein), and monoterpenes, the NO$_x$/BVOC ratios were higher than 8 (nighttime ratios higher than 20) (Figure S3). If
we further consider the major anthropogenic VOCs (toluene, benzene), NO$_x$/VOCs ratios were higher than 5 (nighttime
ratios higher than 10) (Figure S3). The high-NO$_x$ conditions during the campaign were expected to facilitate the oxidation of
aromatic hydrocarbons and the subsequent secondary formation of NACs. Other emissions from biomass burning and coal
combustion were also observed to be contributors of organic aerosols during the campaign (Tang et al., 2018), and they
could also be the precursor sources of NACs. Biomass burning episodes occurred during Wangdu campaign, indicating NAC
emissions from biomass burning (Teich et al., 2017; Tham et al., 2016), which explain the high NAC levels in summer in
Wangdu. The NAC concentrations during summer (including this study) are generally lower than those during spring,
autumn or winter, which could be due to stronger contributions from combustion sources (e.g., biomass burning and coal
combustion) during spring, autumn or winter than those during summer (Chow et al., 2016; Wang et al., 2018b; Kitanovski
et al., 2012; Kahnt et al., 2013).
The NAC compositions are shown in the inserted pie chart in Figure S2. 4-Nitrophenol and 4-nitrocatechol were the
most abundant ones among all the quantified NAC species, accounting for 32.4 % and 28.5 % of the total quantified NACs,
followed by methyl-nitrocatechols (4M5NC, 3M5NC and 3M6NC, 16.2%), methyl-nitrophenol (2M4NP and 3M4NP, 15.6%)
and dimethyl-nitrophenol (8.3%) (Table 1). The contribution of NP and NC could be larger when considering both gas- and
particle-phases. The average concentration of 4NC in both gas- and particle-phases was estimated 2.2 ng/m$^3$ using
equilibrium absorption partitioning model. The total concentration of 4NP (13 ng/m$^3$) in both gas- and particle-phases was
approximated using the measured $F_p$ (17%) by FIGAERO-CIMS (Le Breton et al., 2018). Nitrophenols and nitrocatechols
were generally reported among the most abundant NAC species in previous studies (Table S2 and the references therein).
Nitrophenols could be formed via the oxidation of anthropogenic VOCs (e.g., benzene) in the presence of NO$_2$ and
nitrocatechols were found to dominate the composition of NAC products formed in benzene/NO$_x$ system in laboratory
studies (Xie et al., 2017; Harrison et al., 2005; Yuan et al., 2016; Sato et al., 2007; Ji et al., 2017; Olariu et al., 2002). Thus, it
is not a surprise to observe the high concentrations of nitrophenol and nitrocatechol in the typical high-NO$_x$ and
anthropogenic VOCs dominated environments in summer in Beijing.
The contribution of NP among the total NACs at Changping was higher than that in summer in Hong Kong, while that
of MNC was lower (Table S2 and the inserted pie charts in Figure S2). This NAC composition difference between
Changping and Hong Kong may be a result of different formation pathways for NPs and MNC and different environmental
conditions at the two sites. The gas-phase oxidation of aromatic hydrocarbons (e.g., phenol, benzene) in the presence of NO$_2$
is a major source of NPs (Harrison et al., 2005; Yuan et al., 2016; Sato et al., 2007; Ji et al., 2017; Olariu et al., 2002), while
aqueous-phase oxidation represents the important formation pathway for atmospheric MNC (Frka et al., 2016; Vidovic et al.,
2018). The ambient RH in Hong Kong (>70%) was significantly higher than that in summer in Beijing (5-81%, 37% on
average), thus the relative contribution of aqueous-phase pathways could be more dominant in Hong Kong, promoting the
aqueous-phase formation of MNC. The influence of ambient RH on NAC formation will be further discussed in Section 3.4.
In comparison, more abundant gas-phase formation of nitrophenol was expected in summer in Beijing, under higher
anthropogenic VOCs, high NO$_x$ and low RH conditions. In addition, the lower temperature in summer in Changping was
more favorable for the partitioning of nitrophenols from gas phase into particle phase.
**3.2 Temporal variations and sources of NACs**
Temporal variations of the total quantified NAC concentrations are shown in Figure 3, along with particulate organics,
nitrate, potassium ion, toluene, benzene, acetonitrile, wind speed and RH. During the field campaign, four pollution episodes
(episodes I, II, III, IV), marked by grey shading in Figure 3, were identified through observation of elevated organic aerosols.
Elevated NAC concentrations were observed during pollution episodes, coinciding with the increasing of toluene, benzene,

acetonitrile and potassium. The correlations between NACs and other chemical components are shown in Table S3. The potassium ion was employed to indicate particulate emissions from biomass burning. As the biomass burning-derived immediate VOC precursors to NACs were not detected in this study, acetonitrile was used to track the variations of VOCs released by biomass burning. It was noticed that NACs showed stronger correlations with toluene (r=0.70**), benzene (r=0.64**) or acetonitrile (r=0.61**) than with potassium (r=0.49**). The "**" following the numerical value denotes significant correlation at the 0.01 level. This appeared to suggest that the $NO_x$ oxidation of anthropogenic VOCs and precursor VOCs from biomass burning was a more important source of NACs than primary biomass burning emission in summer in Beijing. A lower correlation between particulate NACs and EC (Table S3, r=0.39**) was also in agreement with the suggestion of the less importance of primary emissions to NACs during the campaign. We note that only particulate NAC concentrations were used to do the correlation analysis. Two atmospheric processes, namely photolysis and gas-to-particle partitioning, could influence the abundance of particle-phase NACs, especially for NP and MNPs, since majority of them was expected to be in the gas phase (Table S1). As such, correlations of particle-phase NP and MNPs with other species may less reliably reflect the underlying associations with the correlation species. As for the relative importance of anthropogenic VOCs and biomass burning-derived VOCs, we do not have direct field measurement data for the differentiation. However, previous studies suggested that the sources of anthropogenic VOCs in summer in Beijing were dominated by vehicle emissions (>50%), with minor contributions from solvent evaporation and biomass burning (Wang et al., 2014; Liu et al., 2008). Modelling studies incorporating emission inventories of the relevant VOC precursors could address this issue and are suggested in future investigation of NAC sources. We note that biomass burning could often be of an anthropogenic origin. Within this work, the term "anthropogenic VOCs" does not include VOCs from human-caused biomass burning activities.

To further investigate the formation of NACs, we examined the time series and day-night variations of individual NAC species (Figures 4, S4 and S5). Daytime enhancements of 4NP, 2M4NP and DMNP, nighttime enhancements of 3M4NC and 4M5NC were observed, and other NAC species did not show discernible day-night variations (Figures 4, S4 and S5), indicating different formation pathways among NAC species. Good inter-species correlations were observed among nitrophenol and its derivatives (2M4NP, 3M4NP, DMNP, r=0.56-0.88), as well as among nitrocatechol and its derivatives (3M6NC, 3M5NC, 4M5NC, r=0.49-0.84). This signaled that the formation and loss pathways as well as the influence factors were likely similar within NP and NC groups. In comparison, the correlations of NACs across the two groups, i.e., between nitrophenol derivatives (MNP, DMNP) and nitrocatechol derivatives (MNC, r=0.05-0.45), were lower (Table S3), suggesting different formation pathways and influence factors. NC and its derivatives showed stronger correlations with toluene, benzene, acetonitrile and $K^+$, compared with NP and its derivatives (Table S3). This was more likely associated with the fact that particle-phase NPs only account for a minor part of the atmospheric NP abundance due to the high vapor pressure of NPs (Table S1). The abundance of particulate NP could largely depend on gas-to-particle partitioning, which is strongly affected by temperature, as well as their gas-phase loss pathways (e.g., photolysis) (Bejan et al., 2007; Yuan et al., 2016;

Sangwan and Zhu, 2018). NC and MNC were mainly present in the particle phase (Table S1). The oxidation degradation rates and photolysis of NC and MNC were therefore much lower. A recent laboratory study found that OH uptake by MNC particles was suppressed by a factor of 4 at RH 15-30% in comparison with dry condition, as a result of competitive co-adsorption of water molecules that occupied reactive sites (Slade and Knopf, 2014). During the campaign, the ambient RH was 37%. Such an RH condition rendered that the OH uptake by particles was suppressed and therefore heterogeneous oxidation of MNC was likely not important.

Nighttime enhancements of 4M5NC and 3M5NC were observed during the whole observation period (Figure 4). A strong correlation between 4M5NC and 3M5NC and their similar temporal variations likely indicated shared similarity in their formation pathways. Previous studies suggested that aqueous-phase oxidation (including photooxidation and nighttime oxidation) is an important formation pathway for atmospheric MNC, especially in polluted high-$NO_x$ environments and relatively acidic particles (pH around 3) (Vidovic et al., 2018; Frka et al., 2016). 4M5NC and 3M5NC showed relatively stronger correlations with RH compared with other NAC species (Table S3), implying the importance of water in their formation processes and the aqueous-phase pathway. During the campaign, the acidic particles (a pH in the range of 2.0-3.7) and the high-$NO_x$ conditions (Wang et al., 2018c; Wang et al., 2018a) provided suitable environments for the aqueous-phase oxidation formation of MNC. The nighttime enhancements of 4M5NC and 3M5NC were more obvious during episode I than during episodes II-IV (nighttime/daytime concentrations at 1.9-3.1 vs. 0.9-1.5) (Figure 4), which suggested that nighttime aqueous-phase formation pathways played more important roles during the first episode. The daytime correlations between 4M5NC or 3M5NC and RH or $NO_2$ were stronger than the nighttime ones (Table S4). The aqueous-phase $NO_x$ oxidation could be more dependent on ambient RH and $NO_2$ levels during the daytime, due to the lower RH and $NO_2$ concentrations than those at night (Figures 3, S3). MNCs also showed good correlations with acetonitrile and potassium, as MNCs could also be formed via the oxidation of biomass burning-derived VOC precursors (e.g., cresol) (Iinuma et al., 2010; Finewax et al., 2018; Olariu et al., 2002). 3M6NC (or 3M4NC isomer) showed different temporal variations from 4M5NC or 3M5NC (Figures 4, S4) and their correlations were lower than that between 4M5NC and 3M5NC (Tables S3, S4), possibly suggesting different formation pathway for 3M6NC (or 3M4NC isomer) from those of 4M5NC or 3M5NC. Quantum calculations have predicted the formation of 3M5NC via aqueous-phase electrophilic substitution and nitration by $NO_2^+$, while the formation of 3M6NC was negligible due to higher activation barriers for nitration of 3-methylcatechol to form 3M6NC (Frka et al., 2016). A dominant presence of 3M5NC in ambient aerosols was also expected according to theoretical predictions (Frka et al., 2016). The 3M5NC concentration was higher than that of 3M6NC in summer in Beijing, consistent with the suggestion from computation study by Frka et al. (2016).

Different from the nighttime enhancements of 4M5NC and 3M5NC, 4NP, 2M4NP and DMNP showed daytime enhancements during the whole campaign (Figures 4, S5). Previously, Yuan et al. (2016) also suggested the daytime gas-phase oxidation of aromatics could represent the major source of NPs, while the contribution from nighttime reaction of

phenol with $NO_3$ radicals was relatively lower (Yuan et al., 2016). The daytime enhancements of NP and its derivatives
(2M4NP, DMNP) were more prominent during episodes II-IV than episode I (daytime/nighttime concentrations at 3.1-4.5 vs.
1.8-2.0) (Figure 4), which indicated that gas-phase photochemical oxidation plays a more important role during the later
period of the campaign. We did not find good correlation between 4NP and $NO_2$ when considering the whole campaign
period (Table S3), while good correlations were observed when treating the daytime and nighttime conditions separately
(Table S4). The strong correlations between 4NP and benzene, toluene or $NO_2$ during daytime and nighttime indicated its
formation via oxidation of benzene and toluene in the presence of $NO_2$ (Table S4). The formation mechanisms of nitrophenol
were different during daytime (OH-initiated photooxidation of aromatics in the presence of $NO_2$) and nighttime
($NO_3$-initiated oxidation of aromatics) (Harrison et al., 2005; Yuan et al., 2016; Sato et al., 2007; Ji et al., 2017; Olariu et al.,
2002), thus the role and influence of $NO_2$ on NAC formation were different. For DMNP, 2M4NP and 3M4NP, they also
showed good correlations with benzene, toluene and $NO_2$ during daytime, but the correlations were absent at night. Instead,
their correlations with RH were higher at night, implying the possible formation via aqueous-phase pathways.
**3.3 The $NO_2$ control of NACs formation**
The analysis in section 3.2 suggests that $NO_x$ oxidation of anthropogenic VOC precursors represented the dominant
sources of NACs in summer in Beijing. To further investigate the impacts of $NO_2$ on NAC secondary formation, we plot the
concentrations of NACs, nitrate ($NO_3^-$) and the $NO_3^-$/NAC ratios as a function of $NO_2$ levels (Figure 5). The variation of
($NO_3^-$)/NACs ratios was employed to illustrate the relative abundance of inorganic nitrate and oxidized organic nitrogen. The
variations during daytime and nighttime were separately considered due to the different atmospheric conditions and
oxidation mechanisms.
Generally, higher concentrations of NACs and nitrate were observed with elevated $NO_2$ concentration levels, in a
nonlinear fashion (Figure 5). During the daytime, NACs increased with $NO_2$, and $NO_3^-$ concentrations and ($NO_3^-$)/NACs
ratios were lower at low-$NO_x$ conditions ($NO_2 < 20ppb$). As $NO_2$ increased to higher than 20 ppb, NAC concentrations did
not increase with $NO_2$ anymore, signaling the transition from $NO_x$-sensitive to $NO_x$-saturated regimes for NAC secondary
formation. At the same time, the $NO_3^-$ concentrations and ($NO_3^-$)/NACs ratios showed increasing trends compared with those
under low-$NO_x$ conditions ($NO_2 < 20ppb$) (Figure 5a, b, c). It was likely that the daytime $NO_2$ was in excess for the oxidation
of ambient VOCs and the NAC formation at $NO_2 > 20$ ppb. Then the excess $NO_2$ would be oxidized to form inorganic nitrate,
producing a shift of products from organic- to inorganic-dominated conditions. Similarly, during nighttime a transition was
observed at $NO_2 \sim 25$ ppb in which oxidation products were shifted from organic- to inorganic-dominance (Figure 5d, e, f).
At $NO_2 > 25$ ppb, the nighttime NAC formation became independent of $NO_2$ concentrations and inorganic nitrate dominated
the $NO_x$ oxidation products. The simplified mechanisms and schematic diagram of the competing formation of inorganic
nitrates and NACs are shown in Figure S6. The nighttime $NO_2$ transition value ($\sim 25$ ppb) was higher than the daytime one
(~20 ppb). The higher concentrations of anthropogenic VOC precursors (Figure S3) and different oxidation mechanisms
(Figure 1) were the potential reasons for elevated $NO_2$ transition value at night.
The compositional variation of inorganic nitrate and NACs described in this work serves as an example in illustrating
that the transition from low- to high-$NO_x$ regimes and the corresponding oxidation products shifting from organic- to
inorganic-dominated conditions exist in polluted urban atmospheres that are characterized by high $NO_x$ and anthropogenic
VOCs. However, the mechanisms as well as transition thresholds were less understood compared with the well-known
BVOCs/$NO_x$ atmospheres. More comprehensive investigation in urban atmospheres is needed to develop more quantitative
understanding of the $NO_x$ regime transition. As only a limited number of VOC species were measured in this study, the $NO_x$
regime transition value was expressed by $NO_2$ concentrations rather than $NO_2$/VOC or $NO_x$/VOC ratios. We also note that
the $NO_x$ regime transition values in other atmospheres could be quite different. The $NO_x$ regime transition values deserve
further investigation through comprehensive lab simulation and field observations to seek a more robust parameter that can
be applied to various atmospheric environments.
The analysis in the previous section indicates that the formation pathways of different NAC species vary from each
other; thus the role and influence of $NO_2$ on their formation are different. The NAC compositions under similar $NO_2$
concentration levels were averaged, with a bin size of 10 ppb $NO_2$. The variation of NAC compositions as a function of $NO_2$
levels is shown in Figure 6 to investigate the influence of $NO_2$ on NAC compositions. The contributions of NCs (standard
deviation< 12% within each $NO_2$ bin) increased and those of NPs (standard deviation< 12% within each $NO_2$ bin) decreased
at elevated $NO_2$ concentrations. The NAC composition remained relatively constant at $NO_2$ >20 ppb, which was
approximately the transition value from low- to high-$NO_x$ regimes. The role of elevated $NO_2$ in promoting formation of NCs
was more obvious than that for NPs. The oxidation of aromatics (e.g., benzene, toluene and VOCs emitted from biomass
burning) in the presence of $NO_2$ represents the major formation pathway of NCs. The formation of NCs would increase with
increasing of ambient $NO_2$. As particle-phase NP and MNP were strongly dependent on the gas-to-particle partitioning and
gas-phase loss (e.g., photolysis), their increasing trends as a function of $NO_2$ were not as obvious as those of NC and MNC.
**3.4 Other influence factors on NACs formation**
Nitration of aromatic hydrocarbons (e.g., benzene and toluene) represents the major source of NACs in summer in
Beijing. NACs generally increased with the increasing of anthropogenic toluene and benzene (Figure 7). During daytime,
when toluene was higher than 0.6 ppb and benzene higher than 0.4 ppb, the NACs concentrations did not increase further
with VOC concentrations (Figure 7a, b). It was likely that toluene or benzene was in excess and the NAC formation became
independent of these precursors. Similarly, the nighttime formation of NACs would become insensitive to these precursors
when toluene was higher than 1 ppb and benzene higher than 0.6 ppb (Figure 7c, d). The transition value of toluene or
benzene was higher at night than during the daytime. This could be due to the significantly higher $NO_2$ levels (significant at
p= 0.01 level) (Figure S3), with higher capacity to oxidize VOC precursors, and different oxidation mechanisms at night.
Though the total NACs didn't show good correlations with ambient RH, good correlations between 3M4NC, 4M5NC
and RH were observed (Table S3, Figure 8). Nitrophenols and methyl-nitrophenols, dominated by gas-phase formation
pathways, were less affected by ambient RH. Aqueous-phase oxidation represented the major formation pathway of 3M4NC
and 4M5NC during the campaign, based on the analysis in section 3.2 and previous studies (Vidovic et al., 2018; Frka et al.,
2016). Elevated ambient RH would favor the water uptake of aerosols and decrease the aerosol viscosity, which favors the
uptake of organic precursors or other gas molecules into the particles, mass diffusion of reactants, and chemical reactions
within the particles (Vaden et al., 2011; Booth et al., 2014; Renbaum-Wolff et al., 2013; Shrestha et al., 2015; Zhang et al.,
2015), and thereby enhance the formation of 3M4NC and 4M5NC in aqueous phase.
The (3M4NC+ 4M5NC)/4NP mass concentration ratios were employed to indicate the relative contribution of
aqueous-phase and gas-phase pathways to NAC formation. The variations of (3M4NC+ 4M5NC)/4NP ratios as a function of
ambient RH during daytime and nighttime are shown in Figure 9. During daytime, this ratio increased with RH when
RH<30%, indicating elevated contribution of aqueous-phase pathways to NAC formation with higher RH conditions. The
ratio remained stable at RH>30% during both daytime and nighttime, suggesting the relative contribution of aqueous-phase
and gas-phase pathways would not increase further with increasing RH beyond RH > 30% (Figures 9a and 9b). The ratio
during the nighttime was obviously higher than during the daytime, indicating that the aqueous-phase oxidation played more
important roles for NAC formation at night. The results implied the importance of aqueous-phase oxidation for the
secondary formation of oxidized organic nitrogen at elevated ambient RH. Due to the limited sample number obtained by
filter-based analysis in this study, the influence of RH or aerosol liquid water content on NAC formation needs further
confirmative investigation using controlled laboratory studies.
The NAC concentrations also showed good correlations with aerosol surface area (Figure 8b). Higher aerosol surface
area would facilitate the partitioning of gas-phase NAC products or precursors into particle phase and the aqueous-phase or
heterogeneous oxidation processes (Kroflic et al., 2015; Bauer et al., 2004; Fenter et al., 1996; Vidovic et al., 2018).
Photolysis is an important loss pathway of NACs and could be the dominant sink for nitrophenols in the gas phase (Bejan et
al., 2007; Yuan et al., 2016). The highest value of $J(O^1D)$ of each day was used to roughly represent the photolysis intensity.
The daytime NAC concentrations showed negative correlations with $J(O^1D)$ (Figure 8c, Table S4), suggesting photolysis as
an important sink for NACs during the daytime.

## 4    Conclusions

Nitroaromatic compounds (NACs) measurements from an intensive field campaign conducted in summer in Beijing

were examined to investigate the abundance and formation characteristics of NACs under high-NO$_x$ and anthropogenic VOCs dominated atmosphere. The average concentration of eight quantified NACs was 6.63 ng/m$^3$, generally higher than those reported in other summertime studies elsewhere. Among the eight NACs, 4-nitrophenol (32.4%) and 4-nitrocatechol (28.5%) were the most abundant, consistent with previous studies, and followed by methyl-nitrocatechol, methyl-nitrophenol and dimethyl-nitrophenol.

Our analysis indicates that the secondary formation via oxidation of anthropogenic VOC precursors (e.g., toluene, benzene) in the presence of NO$_2$ represented a more important source of NACs than primary biomass burning emissions in summer in Beijing. We also observed a transition of oxidation products from organic- to inorganic-dominated conditions as NO$_x$ shifted from low- to high-NO$_x$ regimes. The transition occurred at NO$_2$ of ~20 ppb for the daytime and ~25 ppb for the nighttime atmosphere. Under low-NO$_x$ conditions, NACs were observed to increase with NO$_2$, and the NO$_3^-$ concentrations and (NO$_3^-$)/NACs ratios were lower. Under high-NO$_x$ conditions, the NAC concentration did not further increase with NO$_2$ while the NO$_3^-$ concentrations and (NO$_3^-$)/NACs ratios would show increasing trends. The shift in relative abundance of inorganic nitrate and NACs observed in this work serves as an example in illustrating the demarcation of the low- and high-NO$_x$ regimes in the anthropogenic VOCs-NO$_x$ interacted conditions in polluted urban atmospheres and indicates that NO$_2$ plays important roles in the formation of NACs. The reaction mechanisms are, however, still unclear and deserve further laboratory and field investigation in future studies.

Different day-night variations were observed between the two sub-groups of NACs (i.e, nitrophenols and nitrocatechols). Obvious nighttime enhancements of 3M4NC and 4M5NC and daytime enhancements of 4NP, 2M4NP and DMNP were noted, indicating their different formation pathways. The aqueous-phase oxidation pathways are presumed to be important for the formation of 4M5NC and 3M5NC, under the conditions with high NO$_x$ concentrations and acidic particles during the campaign. Photo-oxidation of toluene and benzene in the presence of NO$_2$ were more important for the formation of nitrophenols. Subsequently, the (3M4NC+ 4M5NC)/4NP mass ratio was employed to probe the relative contribution of aqueous-phase and gas-phase pathways to NAC formation. This ratio would initially increase with RH and remain relatively consistent at RH> 30%, indicating elevated contribution of aqueous-phase pathways to NAC formation under higher RH conditions. Aqueous-phase pathways played more important roles in NAC formation at night than during the daytime.

VOC precursors, aerosol surface area and photolysis were also important factors influencing the NAC formation. NACs generally increased with the increasing of toluene and benzene, implying nitration of aromatic hydrocarbons (e.g., benzene and toluene) may represent the major secondary source of NACs in our study location. The NAC formation would become independent of toluene and benzene, when the daytime concentrations were higher than 0.6 and 0.4 ppb, or the nighttime ones higher than 1 and 0.6 ppb. In addition, aerosol surface area was also an important factor promoting the NAC formation and photolysis could be an important loss pathway of nitrophenols during the daytime.

*Data availability*. The data presented in this article are available from the authors upon request (minhu@pku.edu.cn).

**The Supplement related to this article is available online**

*Author contributions*. MiH, MaH, and SG organized the field campaign. YJW and YCW conducted the offline analysis and analyzed the data. YJW wrote the manuscript with input from JY. All authors contributed to the measurements, discussing results and commenting on the manuscript.

*Competing interests*. The authors declare that they have no conflict of interest.

*Acknowledgements*. This work was supported by National Natural Science Foundation of China (91544214, 91844301, 41421064, 51636003); National research program for key issues in air pollution control (DQGG0103); National Key Research and Development Program of China (2016YFC0202000: Task 3); Hong Kong Research Grant Council (16212017); bilateral Sweden-China framework program on 'Photochemical smog in China: formation, transformation, impact and abatement strategies' by the Swedish Research council VR under contract (639-2013-6917); Project funded by China Postdoctoral Science Foundation (2019M650354).

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

**Figures**

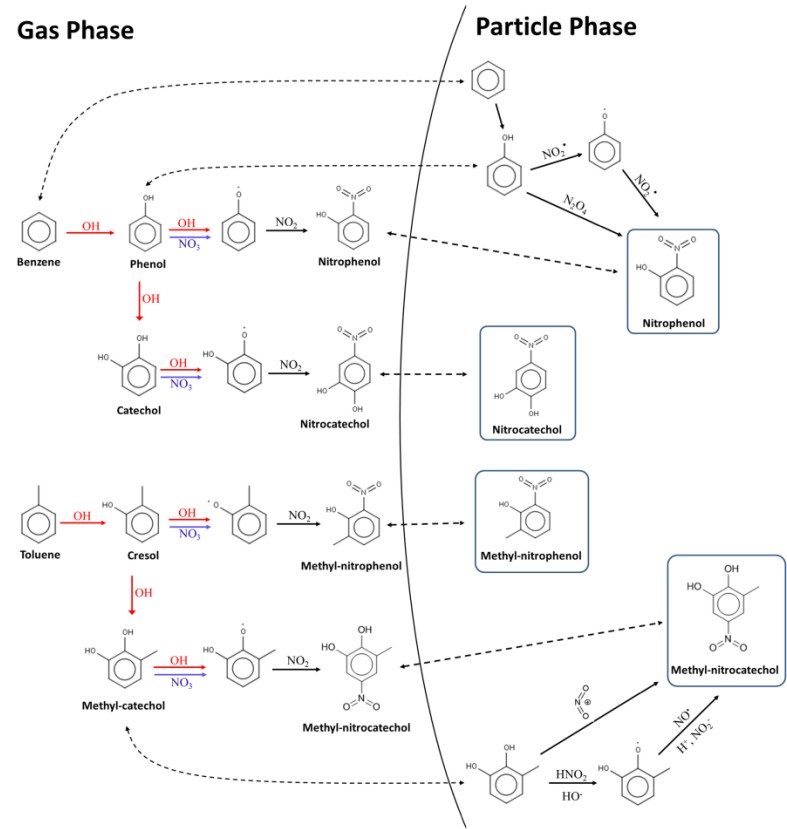


Figure 1 Schematic presentation of NAC secondary formation pathways via the oxidation of benzene, toluene, phenol and
methycatechol in the gas phase and particle phase (Jenkin et al., 2003; Frka et al., 2016; Vione et al., 2004; Vione et al., 2001;
Vidovic et al., 2018).

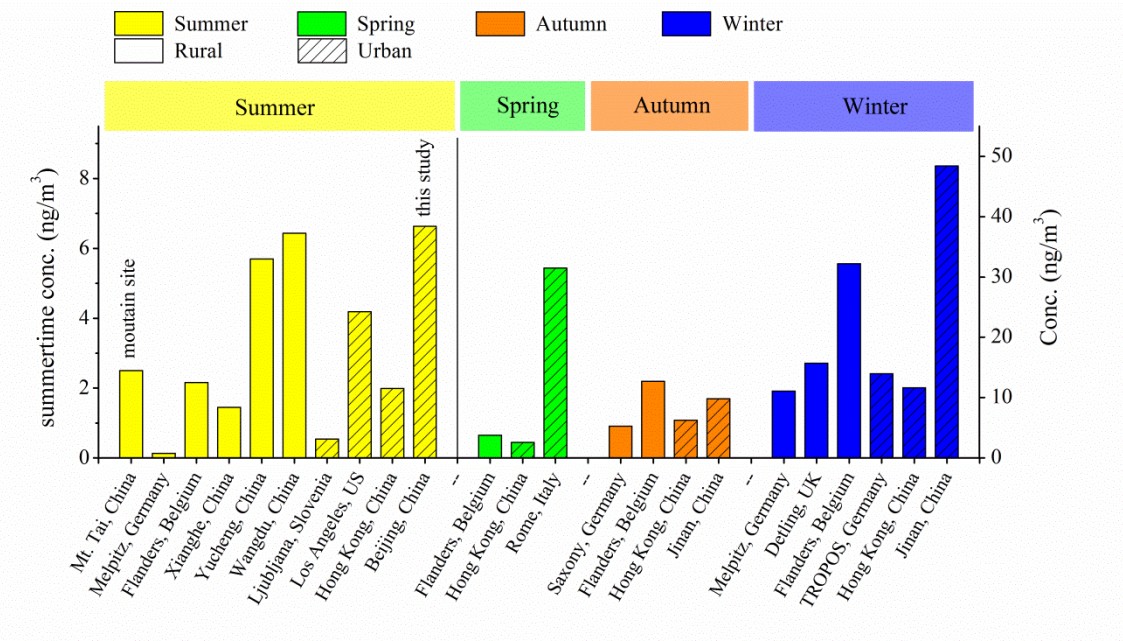


Figure 2 Summary of NAC concentrations across this and prior studies (see Table S2 for the data and references therein).
The NAC concentrations in summer correspond to the left axis and other seasons correspond to the right axis.

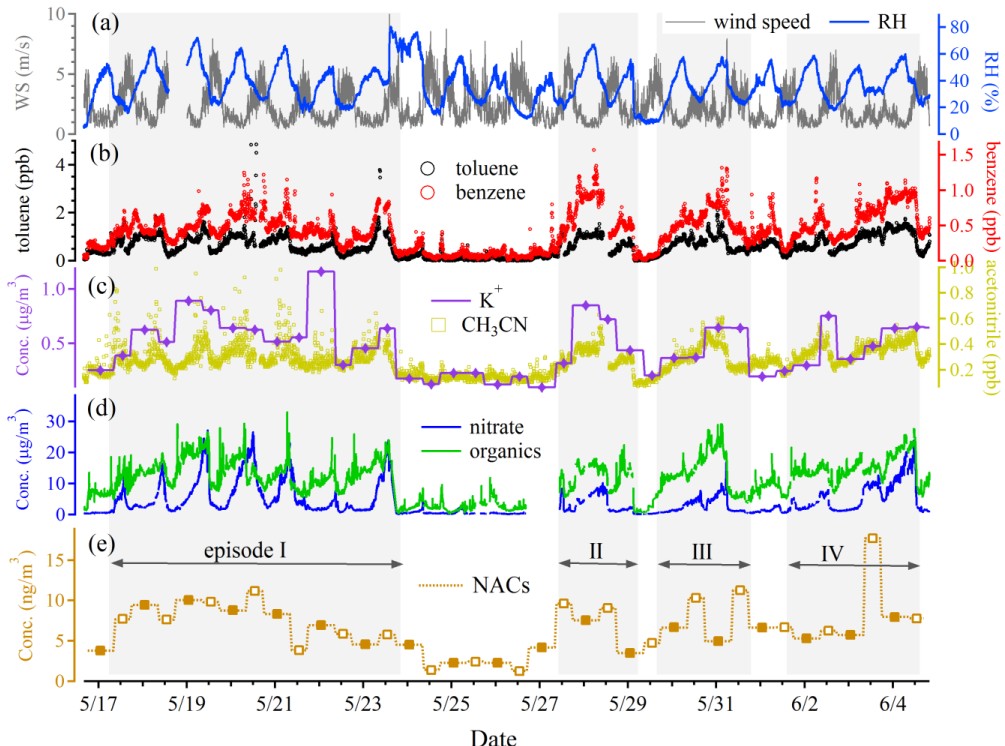


Figure 3 Time series of (a) wind speed (WS) and relative humidity (RH), (b) benzene and toluene, mass concentrations of (c)
K⁺, (d) organics and nitrate, and (e) NACs. The pollution episodes, with elevated organic aerosols, are marked by gray
shading.

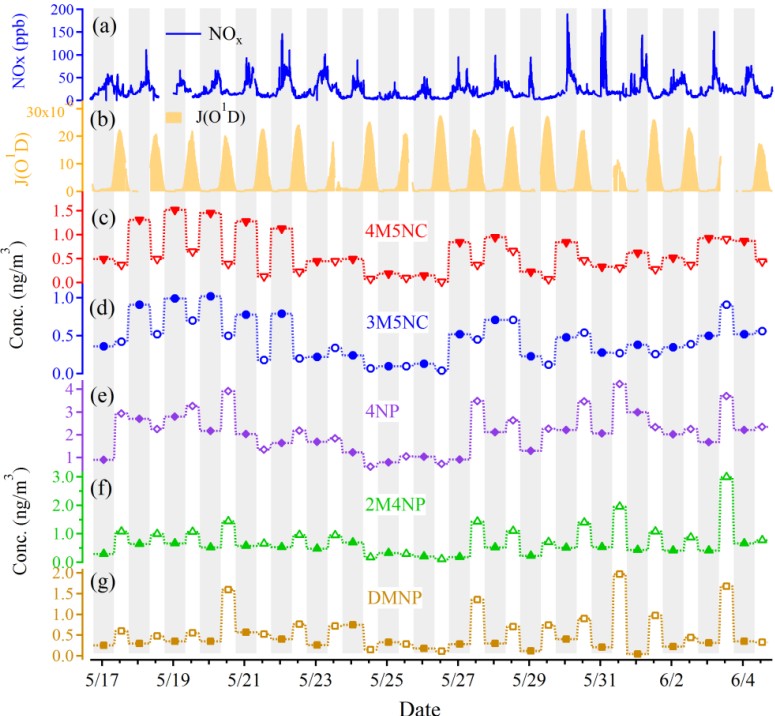


Figure 4 Time series of (a) NOₓ, (b) J(O¹D), (c) 4-methyl-5-nitrocatechol (4M5NC), (d) 4-methyl-5-nitrocatechol (3M5NC),
(e) 4-nitrophenol (4NP), (f) 2-methyl-4-nitrophenol (2M4NP), and (g) dimethyl-nitrophenol (DMNP). The gray background
denotes the nighttime and white background denotes the daytime.

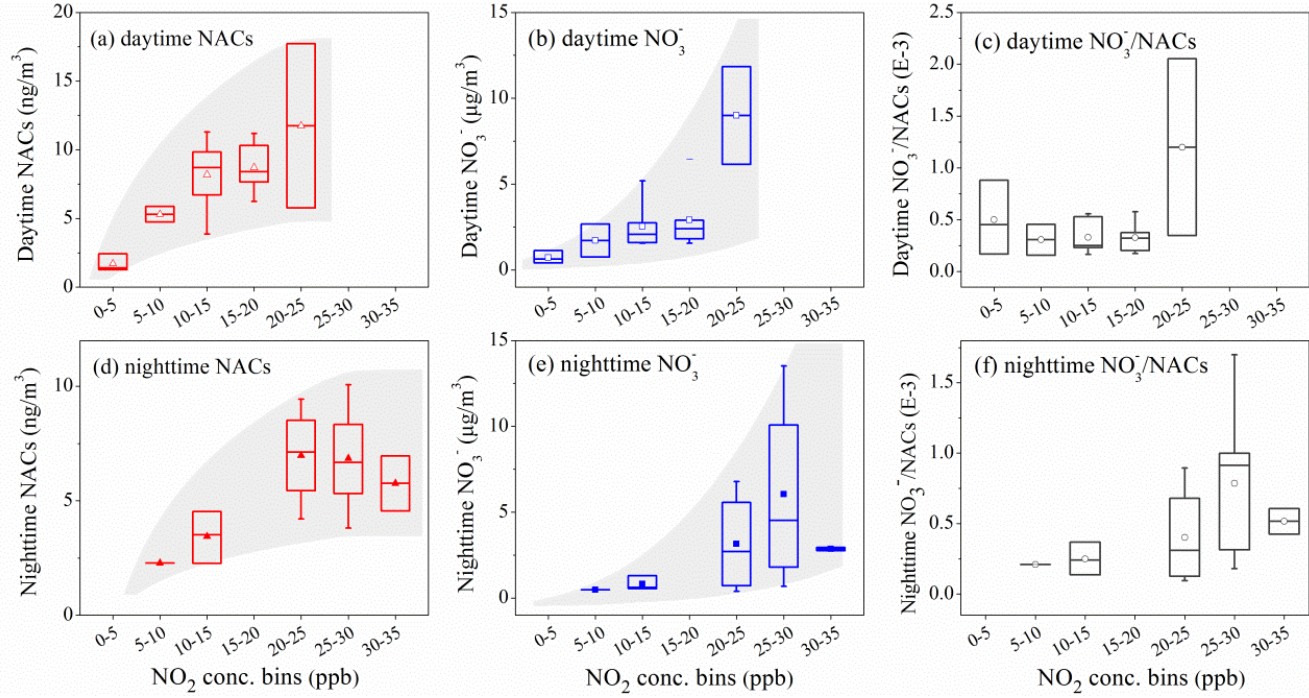


Figure 5 Concentration of NACs, nitrate and $NO_3^-$/NAC ratios as a function of $NO_2$ concentration bins during daytime and
nighttime. The markers represent the mean values and whiskers represent 25 and 75 percentiles.

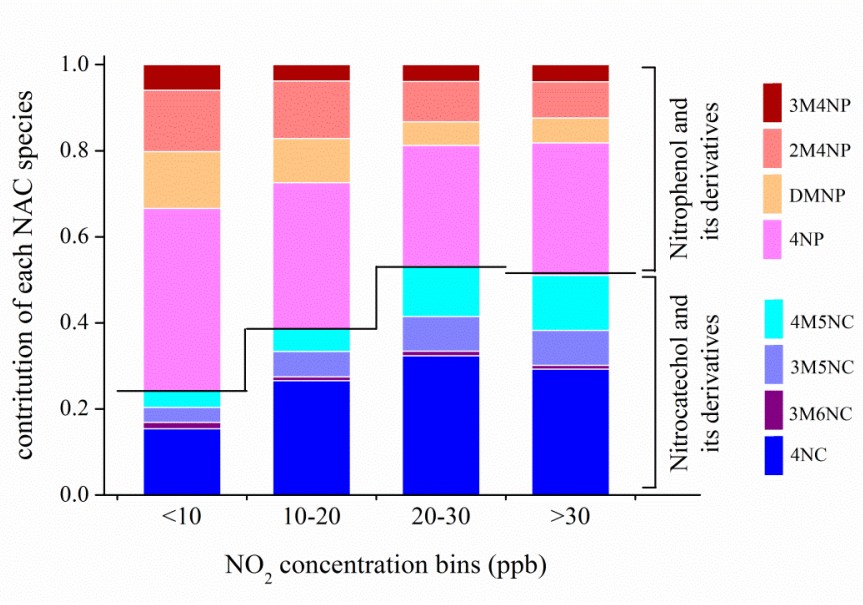


Figure 6 Variation of NAC composition as a function of $NO_2$ concentration bins.


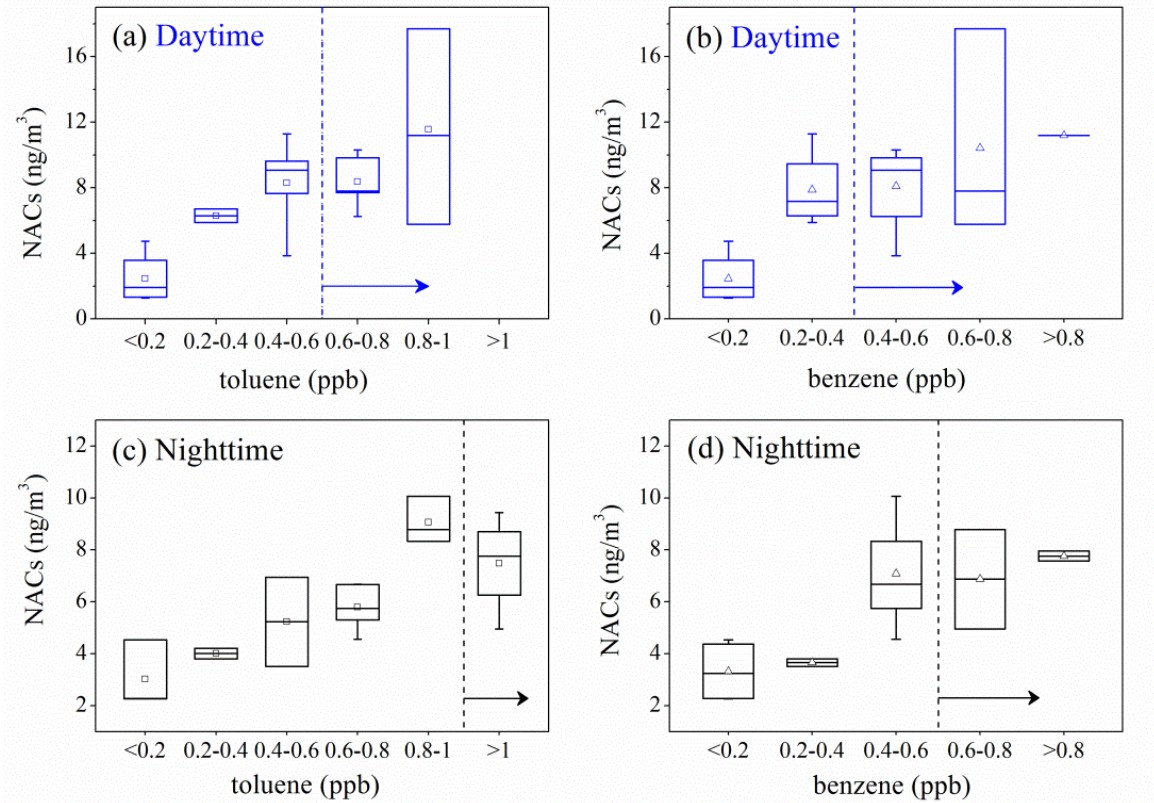


Figure 7 Concentration of NACs as a function of toluene and benzene concentration bins during daytime and nighttime. The
markers represent the mean values and whiskers represent 25 and 75 percentiles.


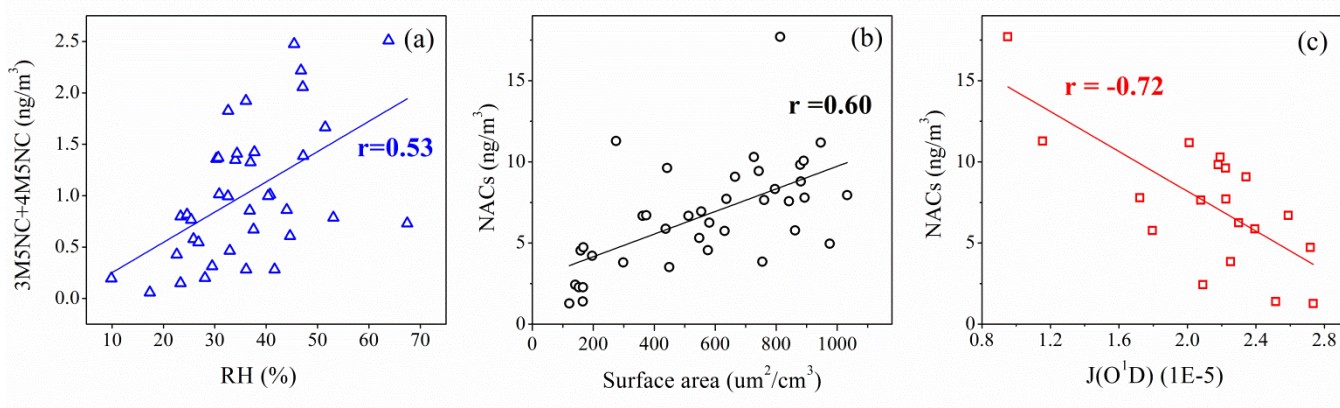


Figure 8 Correlation analysis (a) between (3M5NC+4M5NC) and RH, (b) between NACs and aerosol surface area, (c)
between NACs and J(O$^1$D).

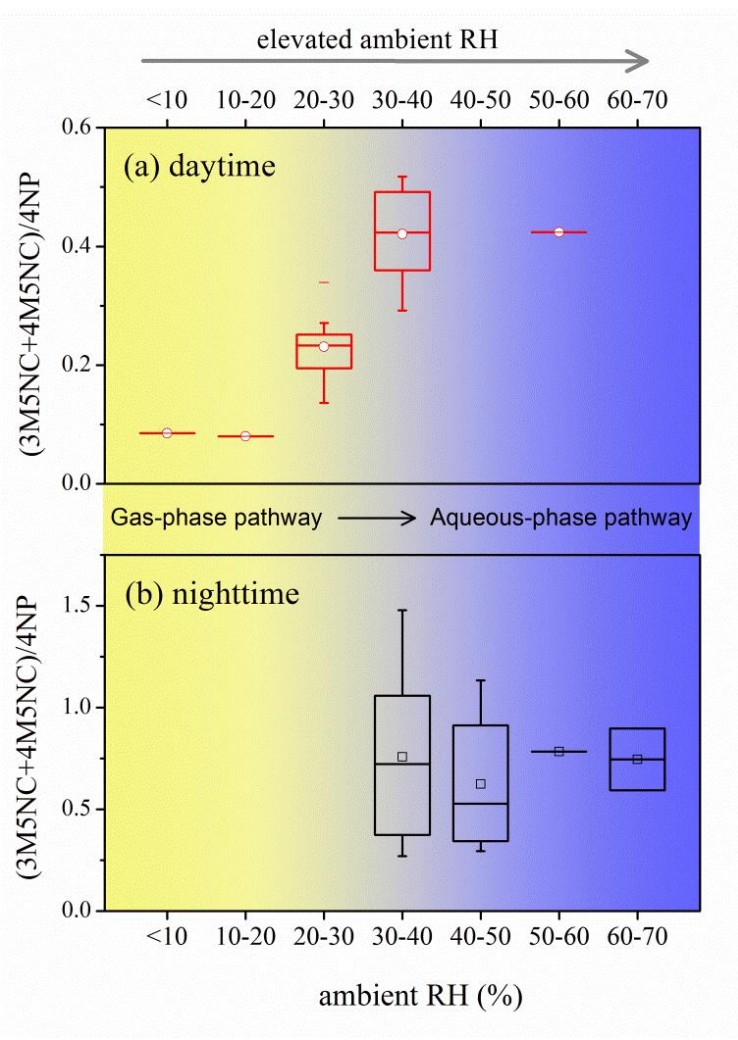


Figure 9 (3M5NC+4M5NC)/4NP concentration ratio as a function of ambient RH during (a) daytime and (b) nighttime.





**Tables**
Table 1 The quantified nitro-aromatic compounds in this study

| compounds | formula | [M-H]$^-$ | retention time (min) | standard | structure | range | average (n=38) |
|---|---|---|---|---|---|---|---|
| 4NP | $C_6H_4NO_3^-$ | 138.02 | 21.3 | 4NP | | 0.60-4.24 | 2.15$\pm$0.93 |
| 3M4NP | $C_7H_6NO_3^-$ | 152.03 | 23.9 | 3M4NP | | 0.08-0.64 | 0.27$\pm$0.12 |
| 2M4NP | $C_7H_6NO_3^-$ | 152.03 | 24.9 | 2M4NP | | 0.11-2.99 | 0.76$\pm$0.55 |
| DMNP | $C_8H_8NO_3^-$ | 166.05 | 26.0, 26.3, 26.9 | 2,6DM4NP | | 0.04-1.97 | 0.55$\pm$0.45 |
| Total NP | | | | | | | 3.72 |
| 4NC | $C_6H_4NO_4^-$ | 154.01 | 18.9 | 4NC | | 0.16-6.89 | 1.89$\pm$1.28 |
| 4M5NC | $C_7H_6NO_4^-$ | 168.03 | 21.8 | 4M5NC | | 0.02-1.52 | 0.56$\pm$0.40 |
| 3M6NC | $C_7H_6NO_4^-$ | 168.03 | 23.2 | 4M5NC | | 0.02-0.19 | 0.07$\pm$0.03 |
| 3M5NC | $C_7H_6NO_4^-$ | 168.03 | 23.5 | 4M5NC | | 0.04-1.02 | 0.44$\pm$0.27 |
| Total NC | | | | | | | 2.96 |
