# Peer review of "The formation of nitro-aromatic compounds under high NOx and anthropogenic VOC conditions in urban Beijing, China"

_Atmospheric Chemistry and Physics, 2018_

## Referee Comment (RC1) · Anonymous Referee #2 · 5 Feb 2019

**Review of acp-2018-1256**

Reviewer 3

February 4, 2019

**1    General Comments**

This paper examines the formation of nitro-aromatic compounds (NACs) in the Beijing summer under high NOx and high anthropogenic VOC conditions. NAC formation is of interest owing to their light- absorbing and toxicological properties. Formation properties of both the nitro-phenols (NPs) and the nitro-catechols (NC) are diurnal with the latter forming at night and the former forming predominantly during the day time. Excess NOx and VOC concentration thresholds were reported. Higher NOx generally favored formation of the NCs. Generally, VOC concentration thresholds of < 1 ppb were observed. It was concluded that aqueous phase oxidation was more important at night than daytime.

The study's subject matter is of interest to the *ACP* audience, and the paper is well written. NACs in biomass burning and in the atmosphere are becoming well investigated, and while fewer studies may examine NACs in the urban environment, a clearer statement about the novelty of the current research is due. For example, Figure 1 shows that NACs have been measured throughout the world, including at several urban sites in China. So, this paper should explicitly state why another study is needed on this topic now. Additionally, the study would benefit from showing the NC and NP compound peaks and their resolution in the LC-MS chromatograms. Next, the pollution events and $K^+$ need to clearly connect to the main story of the paper (e.g. Figure 3). The discussion suggests that biomass burning is mainly a non-factor despite the $K^+$ possibly showing otherwise, and the diurnal profile seems much more robust than any of the four so-called 'pollution events'. In other words, the importance of the individual pollution events and the source of the events is just isn't clear enough. These points should be better addressed during revision. Finally, the paper would also benefit from further analysis of the potential gas-phase concentrations of some of these compounds and quantification of the likelihood of volatilization losses. Use of vapor pressures or partitioning coefficients should help with this issue. Once addressing these concerns and the additional comments below, the paper should be publishable.

**2    Specific comments**

1. lines 19-21: This sentence should be revised for clarity. 'high NOx anthropogenic VOCs' is this one or two concepts? Also, not certain what 'influence factors' are. Please fix.

2. line 21: Is this total concentration an average? Or the top end of the range? Be specific.

3. line 29: just state that there was excess NOx for VOC oxidation. As written the sentence is awkward.

4. line 38: Where are the aerosol surface area data?

5. lines 112-117: Please provide a chromatogram that shows the separation. Many of these compounds are isomers, and it is important to see evidence of sound chromatographic resolution. Evidence of peak quality for this study is warranted.

6. Figure 1: these reaction schematics are not chemically balanced. If -$H_2O$ is removed than it should be indicated in the reaction scheme. What happens to the $NO_3$? This reaction scheme should show the reader exactly what happens chemically. Are these compounds also present in the gas-phase?

7. line 162-165: This last sentence should be revised for clarity. Can't tell which conditions go with which season.

8. lines 173-174: So, it's a high NOx *and* a high anthropogenic VOC environment?

9. lines 179-181: Please clarify the sentence about 'Different NAC compositions'. How much higher is the RH in Hong Kong compared with Beijing in summer?

10. lines 205-209: It is unclear what is being discussed here with regard to the pathway, loss mechanisms, and groups. Please be specific and quantitative if possible.

11. lines 209-210: Why was the gas-phase not sampled for this study? It would have brought more clarity and we could have learned if the mass was simply distributed in the gas-phase as opposed to being 'lost'. It may be a good idea to use individual NAC concentrations, vapor pressures and day and night time temperature to estimate the total mass in both the gas- and particle-phase.

12. line 271: please clarify 'interacted conditions'

13. lines 275-277: Please rewrite this sentence for clarity.

14. Figure 6: While this figure is meant to indicate a general trend, it is unclear where the values in this figure came from. Are these averages? How much error are in these values? Please be specific.

15. lines 289-291: Again, a table containing the vapor pressures for these NAC compounds may help explain some of these observations. Try the CHEMSPIDER web site.

16. line 301: is this statistically 'significant'?

17. 347-348: The last sentence in this paragraph is unclear.

---

## Referee Comment (RC2) · Anonymous Referee #1 · 7 Feb 2019

General comments:

This work measured two groups of nitro-aromatic compounds (NACs), including nitrophenol (NP) and nitrocatechol (NC) like species at an urban site in summer Beijing. Their diurnal variations and formation pathways were discussed thoroughly with additional data of VOCs, meteorological parameters, etc. The authors found that the secondary formation is a more dominant NAC source than biomass burning, and the two types of NACs have distinct formation mechanisms. A concentration ratio of NACs was applied to indicate the relative importance of aqueous-phase and gas-phase oxidation. The NACs are an important group of light-absorbing compounds in ambient aerosols. Their compositions and sources in the atmosphere are very complex and studied intensively in recent years. The topic of this work is very interesting, and the manuscript is very well organized and written. However, some of the conclusions might need more work with more valid evidence.

1. Page 7, lines 194-198.
As NACs were more correlated with toluene and benzene (r = 0.64 – 0.70) than potassium (r = 0.49), the authors concluded that the NACs were more likely from photo-oxidation of VOCs than biomass burning in summer Beijing.
First, please provide the significance level of the correlations.
Second, the authors did regression analysis by comparing the particle-phase NACs with VOCs and potassium. In the atmosphere, NACs are mainly subject to two atmospheric processes, photolysis and gas-to-particle partitioning; toluene and benzene are in gas phase and precursors for SOA. The authors might need to consider the uncertainties in correlations due to atmospheric transformations.

2. Page 7, line 211 – 217
4M5NC and 3M5NC had higher nighttime concentrations than daytime, and were more correlated with RH than other NACs, which were ascribed to the aqueous-phase oxidation.
However, the authors might need to consider other possibilities.
1). Nitrocatechols are easier to be oxidized than nitrophenols. The second-order rate constant for nitrocatechol ($5 \times 10^9$ $M^{-1}$ $s^{-1}$) with OH in aqueous phase is much higher than nitrophenol ($3.7 \times 10^9$ $M^{-1}$ $s^{-1}$) (Hems and Abbatt , 2018). So the higher nighttime concentrations of 4M5NC and 3M5NC might also be associated with their shorter life time during daytime.
2) The nitrocatechol has one more hydroxyl group and is more water soluble than the nitrophenol. Besides gas-to-particle partitioning, the nitrocatechols might also be dissolved into particle water, which could be another explanation for the higher correlations between nitrocatechols and RH%.

3. Page 8, lines 224- 226.
Could it be resulted from their different life times?

4. Page 9, line 250.

Why $NO_X$ oxidation of anthropogenic VOC precursors is the dominant source of NACs? Can we estimate the relative contributions of biomass burning and $NO_X$ oxidation?

5. Page 10, lines 305-306. Page 11, lines 317-319.

As mentioned in comment 2, please consider the shorter life times of nitrocatechols than nitrophenols during the daytime, and the dissolution of nitrocatechols in particulate water as RH% increases.

Moreover (connect to Page 6, lines 181-183), the authors thought that a higher relative humidity (RH%) would increase the relative contribution of aqueous-phase oxidation. Is the aqueous-phase oxidation in ambient aerosols only dominated by RH%?

Specific comments

1. Page 4, line 122. Please provide the recoveries for the target compounds.

**References**

Hems, R. F., and Abbatt, J. P. D.: Aqueous Phase Photo-oxidation of Brown Carbon Nitrophenols: Reaction Kinetics, Mechanism, and Evolution of Light Absorption, ACS Earth and Space Chemistry, 2, 225-234, 10.1021/acsearthspacechem.7b00123, 2018.

---

## Referee Comment (RC3) · Anonymous Referee #3 · 11 Feb 2019

**GENERAL COMMENTS**

Based on the extensive data from the field collected during the summertime campaign in Beijing, the manuscript discusses formation mechanisms of the most abundant groups of atmospheric nitroaromatic compounds, nitrophenols and nitrocatechols. The represented dataset is very valuable and the manuscript is well structured, however the conclusions made by the authors are sometimes vague and require more justification.

The authors distinguish between biomass-burning and anthropogenic emissions. As BB emissions are often of an anthropogenic origin, please make it clear that you mean traffic emissions when referring to anthropogenic emissions.

In the introduction, much space is devoted to biomass burning as a possible source of NP and NC, however the characteristic methoxyphenols (substituted guaiacols and syringols) and their nitrated analogues are not even mentioned. Why don't you report the concentrations of (nitro)guaiacols and (nitro)syringols (maybe you didn't detect them?). This could help support the statement that anthropogenic (traffic?) emissions prevailed over biomass burning emissions during the campaign. To my opinion, this assumption is not justified enough in the manuscript (although it is very likely to be true). Also, explain why primary emissions are not likely the important source of measured NAC.

In the cited paper of Frka et al. it is nicely demonstrated that the usage of surrogate molecules for the quantification of aromatic isomers by LC-MS can result in the over- or underestimation of the concentration of analyte, even for the factor of 10. The case of 3M4NC and 3M6NC is specifically exposed, as well as their misidentification in the past studies. Make it clear that you considered this issues during the analysis (if not, 3M6NC is more likely to be 3M4NC) and comment on the determined concentration of 3M6NC by use of 4M5NC standard. Reconsider also the conclusion in L224-226. Moreover, how do you know that DMNP peaks are truly dimethylnitrophenols and not ethylnitrophenols or their methoxylated isomers?

The language should be edited before the manuscript is published in ACP.

**SPECIFIC COMMENTS**

L60-62: how can MNC form by the oxidation of catechol? Did you mean methylcatechol?

L64-65: the sentence is misleading. In Fig. 1 you only show the most generally accepted pathways of gas-phase NAC formation. Revise the sentence for clarity and/or show also the aqueous-phase chemistry which you are referring to later in this paragraph. I strongly suggest extending the figure to at least the aqueous-phase formation of NC, which you show to be dominant in the atmosphere. Include also the relevant references in the figure caption.

L71-72: refer here to the very recent and comprehensive study on the possible nitration mechanisms of activated methoxyphenols in the atmospheric aqueous phase by Kroflič et al.: https://pubs.acs.org/doi/10.1021/acs.est.8b01903

L72: 'Nitrophenols could form through the nitration and hydroxylation of benzene...' note that NO2 group deactivates the aromate, whereas OH makes it more reactive. Therefore, oxidation-first and nitration-second is more likely to occur. Change the order accordingly.

L93-94: I cannot see it obvious that anthropogenic emissions dominated during the campaign.

L149-150: check again, NP is also not always higher than in other studies.

L192 (and Fig. 3 caption): what kind of pollution episodes?

L210, L290, L326-327: It seems like photolysis is not important for the aqueous-phase products (NC)?

L219-222: refer also to Fig. 9 – at the nightime RH levels, daytime ratios are also not dependent on RH anymore.

L259-260: 'NO3-concentrations and (NO3-)/NACs ratios showed significant increasing trends (Figure 5a, b, c).' Not sure if I can see this, especially for the ratio. Please correct.

L325: Use here the original research of Frka et al. to avoid misunderstanding: https://pubs.acs.org/doi/abs/10.1021/acs.est.5b01811

Fig. 7: cannot find r value referred to in the caption.

Fig. S5: this is again only gas-phase chemistry presented. Make this crystal clear or add also heterogeneous/aqueous-phase pathways.

**TECHNICAL CORRECTIONS**

The usage of word 'obvious' is excessive. You should avoid it throughout the text.

It is not clear to me why you used Results and Discussion section 3 and later on again a Discussion section 4. I would use only section 3 – Results and Discussion.

L16 and L47: the phrase 'ultraviolet light absorption by brown carbon' is incorrect, the characteristics of BrC is visible light absorption. Please correct accordingly.

L162-165: the sentence needs revision.

In general, the language, especially the grammar, requires editing.

---

## Author Comment (AC1) · 19 Apr 2019

Dear co-editor and referees,

We appreciate all your detailed and valuable comments on our manuscript of "acp-2018-1256". Please see the point-by-point response below and changes are marked blue in the revised manuscript.

**Referee #1**

**1 General Comments**

*This paper examines the formation of nitro-aromatic compounds (NACs) in the Beijing summer under high NOx and high anthropogenic VOC conditions. NAC formation is of interest owing to their light- absorbing and toxicological properties. Formation properties of both the nitro-phenols (NPs) and the nitro-catechols (NC) are diurnal with the latter forming at night and the former forming predominantly during the day time. Excess NOx and VOC concentration thresholds were reported. Higher NOx generally favored formation of the NCs. Generally, VOC concentration thresholds of < 1 ppb were observed. It was concluded that aqueous phase oxidation was more important at night than daytime.*

*The study's subject matter is of interest to the ACP audience, and the paper is well written. NACs in biomass burning and in the atmosphere are becoming well investigated, and while fewer studies may examine NACs in the urban environment, a clearer statement about the novelty of the current research is due. For example, Figure 1 shows that NACs have been measured throughout the world, including at several urban sites in China. So, this paper should explicitly state why another study is needed on this topic now. Additionally, the study would benefit from showing the NC and NP compound peaks and their resolution in the LC-MS chromatograms. Next, the pollution events and K+ need to clearly connect to the main story of the paper (e.g. Figure 3). The discussion suggests that biomass burning is mainly a non-factor despite the K+ possibly showing otherwise, and the diurnal profile seems much more robust than any of the four so-called `pollution events'. In other words, the importance of the individual pollution events and the source of the events is just isn't clear enough. These points should be better addressed during revision. Finally, the paper would also benefit from further analysis of the potential gas-phase concentrations of some of these compounds and quantification of the likelihood of volatilization losses. Use of vapor pressures or partitioning coefficients should help with this issue. Once addressing these concerns and the additional comments below, the paper should be publishable.*

**Response**: Thanks for your suggestions.

1) *NACs in biomass burning and in the atmosphere are becoming well investigated, and while fewer studies may examine NACs in the urban environment, a clearer statement about the novelty of the current research is due. For example, Figure 1 shows that NACs have been measured throughout the world, including at several urban sites in China. So, this paper should explicitly state why another study is needed on this topic now.*

**Responses**: Though NACs were investigated throughout the world, including several sites (two rural sites in Hebei Province and an urban site in Hong Kong) in China. Few studies in China has been as comprehensive in supporting measurements as this study. In this work, with the additional field data of inorganic aerosols, VOC precursors, gases and meteorological parameters, we are able to gain more comprehensive understanding of the characteristics, secondary formation and

influence factors of NACs in urban atmospheres characteristic of high $NO_x$ and high anthropogenic VOC concentrations, which was not done in the past studies conducted in urban environment. A statement about the novelty of the current research has been added in lines 83-86.

Lines 83-86: However, few observational field studies have been conducted to investigate the formation of NACs in urban atmospheres. In this work, we report results from an intensive field campaign conducted in summertime Beijing, aiming to gain understanding of ambient concentration variation characteristics of NAC, relative importance of various proposed formation pathways and major influence factors in high $NO_x$ and anthropogenic VOCs dominated urban atmospheres.

2)  *the study would benefit from showing the NC and NP compound peaks and their resolution in the LC-MS chromatograms.*

**Responses**: The chromatograms of NAC standards and an ambient sample are added in Figure S1.

[Figure]

Figure S1 Extracted ion chromatograms for (a) a standard solution containing 4-nitrocatechol (4NC), 4-nitrophenol (4NP), 2-methyl-4-nitrophenol (2M4NP), 3-methyl-4-nitrophenol (3M4NP), 2,6-dimethyl-4-nitrophenol (2,6DM4NP) and internal standard and (b) an ambient sample collected during the campaign.

3)  *the pollution events and $K^+$ need to clearly connect to the main story of the paper (e.g. Figure 3). The discussion suggests that biomass burning is mainly a non-factor despite the $K^+$ possibly showing otherwise, and the diurnal profile seems much more robust than any of the four so-called `pollution events'. In other words, the importance of the individual pollution events and the source of the events is just isn't clear enough. These points should be better addressed during revision.*

**Responses**:

**(1)** In this study, the potassium ion was employed to indicate the particles (particulate NACs) primarily emitted from biomass burning. Gas-phase acetonitrile was employed to indicate the variations of gases (VOC precursors) released by biomass burning. The correlation analysis between NACs and acetonitrile could indicate the secondary formation of NACs via the oxidation of VOCs (e.g. cresol) emitted from biomass burning. Lower correlation between NACs and $K^+$ (r=0.49) than that between NACs and acetonitrile (r=0.61) suggested that primary biomass burning emissions were less important source of NACs than oxidation of VOCs released by biomass burning. The revised text now provides a clearer explanation (lines 243-250).

Lines 243-250: The potassium ion was employed to indicate particulate emissions from biomass

burning. As the biomass burning-derived immediate VOC precursors to NACs were not detected in this study, acetonitrile was used to track the variations of VOCs released by biomass burning. It was noticed that NACs showed stronger correlations with toluene (r=0.70**), benzene (r=0.64**) or acetonitrile (r=0.61**) than those with potassium (r=0.49**). This appeared to suggest that the $NO_x$ oxidation of anthropogenic VOCs and precursor VOCs from biomass burning was a more important source of NACs than primary biomass burning emission in summer in Beijing. A lower correlation between particulate NACs and EC (Table S3, r=0.39**) was also in agreement with the suggestion of the less importance of primary emissions to NACs during the campaign.

**(2)** The nighttime aqueous-phase formation pathways played more important roles during episode I, and the daytime gas-phase photooxidation pathways could be more important during episodes II-IV. Description of the differences of pollution episodes in relation to sources/formation mechanisms was revised to provide more clarity (lines 289-291, 307-310).

Lines 289-291: The nighttime enhancements of 4M5NC and 3M5NC were more obvious during episode I than those during episodes II-IV (nighttime/daytime concentrations at 1.9-3.1 vs. 0.9-1.5) (Figure 4), which suggested that nighttime aqueous-phase formation pathways played more important roles during the first episode.

Lines 307-310: The daytime enhancements of NP and its derivatives (2M4NP, DMNP) were more prominent during episodes II-IV than episode I (daytime/nighttime concentrations at 3.1-4.5 vs. 1.8-2.0) (Figure 4), which indicated that gas-phase photochemical oxidation play more important roles during the later period of campaign.

4) *the paper would also benefit from further analysis of the potential gas-phase concentrations of some of these compounds and quantification of the likelihood of volatilization losses. Use of vapor pressures or partitioning coefficients should help with this issue.*

**Responses**: Thanks for the suggestion. We now have added analysis to provide a semi-quantitative estimate of the gas-phase presence of the NACs. The information of the different gas-particle distributions between two sub-groups of NACs (i.e., nitrophenols and nitrocatechols) is used to better explain their distinct formation mechanisms.

The methodology of estimating gas-phase concentrations of NACs is described in Section 2.4 and copied below for easy reference. New discussions arising from this added information are added (lines 216-219, 250-254, 271-275, 359-361).

**Section 2.4 Estimation of the gas-phase NACs**

The concentrations of gas-phase NACs were not measured in this study. They were calculated based on the measured particle-phase NAC concentrations and equilibrium absorption portioning theory (Pankow, 1994a, b; Pankow et al., 2001) (Eqs. 1, 2):

$$F_p = (1 + \frac{C^*}{C_{OA}})^{-1} = \frac{c_p}{c_g + c_p} \qquad \text{(Eq. 1)}$$

where $F_p$ is the fraction of NACs in the particle-phase. $C_{OA}$ is the concentrations of organic aerosols (OA), calculated to be OC multiplying by 1.6. $c_g$ and $c_p$ are the concentrations of NACs in gas phase and particle phase, respectively. $C^*$ is the effective saturation mass concentration ($\mu g/m^3$), and is calculated using Eq. 2:

$$C^* = \frac{M 10^6 \zeta P_V}{760 RT} \qquad \text{(Eq. 2)}$$

where M is the molecular weight of NACs (g/mol). ζ is the activity coefficient of the species (assumed =1). $R$ is the gas constant (8.314 J/(mol K)), $T$ is the temperature (K), and $P_v$ (Pa) is the saturated pressure. $P_v$ at the average temperature during the campaign (296 K) is calculated using the multiphase system online property prediction tool developed by University of Manchester (UManSysProp, http://umansysprop.seaes.manchester.ac.uk). The vapor pressures were estimated using Nannoolal approach, and the boiling points were estimated using the Joback and Reid approach.

The estimated $P_v$, $F_p$ and gas-phase concentrations of NACs are listed in Table S1. 4NP and methyl-nitrophenols (2M4NP and 3M4NP) were predicted to be mainly in the gas phase ($F_p<10\%$) while DMNP, 4NC and MNC (3M6NC, 3M5NC and 4M5NC) were mainly in the particle phase ($F_p>60\%$). The gas-phase DMNP and MNC ($F_p>95\%$) would not be further discussed in this study. While the equilibrium model gives reasonable estimation of $F_p$ and gas-phase concentrations for nitrocacatechols, it overestimates the vapor pressure of NPs by several orders of magnitude (Bannan et al., 2017). The estimated $F_p$ (0.83%) was obviously lower than the measured values for 4NP. For example, Cecinato et al. (2005) measured $F_p$ of 4NP and 3M4NP to be 82% and 78%, respectively in downtown Rome; Le Breton et al. (2018) reported $F_p$ of nitrophenol at ~17% using Chemical-Ionization Mass Spectrometer (CIMS) coupled with the Filter Inlet for Gases and AEROsols (FIGAERO) during this campaign. We note that CIMS could not distinguish the isomers (e.g. 2NP) of 4NP, however, the measured $F_p$ values showed us the range of particulate fraction of 4NP during the campaign. The equilibrium absorption portioning model could underestimate the $F_p$ of 4NP by ~20 times. Thus, the gas-phase 4NP concentration was roughly calculated using the measured $F_p$ (17%) by FIGAERO-CIMS (Le Breton et al., 2017).

**Table S1** The saturated pressure ($P_v$), NAC fraction in particles ($F_p$) and NAC concentrations in gas-phase

| compounds | $P_v$ (Pa) | $F_p$ (%) | | conc. in gas phase (ng/m$^3$) | |
|---|---|---|---|---|---|
| | | range | average | range | average |
| 4NP | 3.0E-02 | 0.25-1.3 | 0.83 | 143-566 | 268 |
| 2M4NP | 4.2E-03 | 1.2-7.6 | 5.1 | 5.2-42 | 15 |
| 3M4NP | 4.2E-03 | 1.2-7.6 | 5.1 | 1.9-20 | 5.8 |
| DMNP | 5.3E-04 | 99-100 | 100 | 0.000-0.007 | 0.002 |
| 4NC | 3.4E-05 | 67-91 | 85 | 0.06-0.79 | 0.27 |
| 3M6NC | 3.1E-06 | 95-99 | 98 | 0.000-0.003 | 0.001 |
| 3M5NC | 3.1E-06 | 95-99 | 99 | 0.002-0.017 | 0.007 |
| 4M5NC | 3.1E-06 | 95-99 | 98 | 0.001-0.028 | 0.009 |

Lines 216-219: The contribution of NP and NC could be larger when considering both gas- and particle-phases. The average concentration of 4NC in both gas- and particle-phases was estimated 2.2 ng/m$^3$ using equilibrium absorption portioning model. The total concentration of 4NP (13 ng/m$^3$) in both gas- and particle-phases was approximated using the measured $F_p$ (17%) by FIGAERO-CIMS.

Lines 250-254: We note that only particulate NAC concentrations were used to do the correlation analysis. Two atmospheric processes, namely photolysis and gas-to-particle partitioning, could influence the abundance of particle-phase NACs, especially for NP and MNPs, since majority of them was expected to be in the gas phase (Table S1). As such, correlations of particle-phase NP and MNPs with other species may less reliably reflect the underlying associations with the correlation species.

Lines 271-275: This was more likely associated with the fact that particle-phase NPs only account for a minor part of the atmospheric NP abundance due to the high vapor pressure of NPs (Table S1). The abundance of particulate NP could largely depend on gas-to-particle partitioning, which is strongly affected by temperature, as well as their gas-phase loss pathways (e.g. photolysis) (Bejan et al., 2007; Yuan et al., 2016; Sangwan and Zhu, 2018). NC and MNC were mainly present in the particle phase (Table S1).

Lines 359-361: As particle-phase NP and MNP were strongly dependent on the gas-to-particle partitioning and gas-phase loss (e.g. photolysis), their increasing trends as a function of $NO_2$ were not as obvious as those of NC and MNC.

**2 Specific comments**

1.  *lines 19-21: This sentence should be revised for clarity. `high NOx anthropogenic VOCs' is this one or two concepts? Also, not certain what `influence factors' are. Please fix.*

    **Response**:

    (1) We now revise 'High $NO_x$ anthropogenic VOCs' to "High $NO_x$ and anthropogenic VOCs" throughout the manuscript.

    (2) "Influence factors" means "factors that influence NAC formation, including $NO_2$, VOC precursors, RH and photolysis". The relevant text in the abstract has been re-worded to improve clarity (lines 18-19).

    Lines 18-19: We investigated the factors that influence NAC formation (e.g. $NO_2$, VOC precursors, RH and photolysis)…

2.  *line 21: Is this total concentration an average? Or the top end of the range? Be specific.*

    **Response**: It's the average concentration of all the quantified NACs. It has been revised to be specific (lines 20-21).

    Lines 20-21: The average total concentration of the quantified NACs was 6.63 ng/m$^3$, higher than those reported in other summertime studies (0.14- 6.44 ng/m$^3$).

3.  *line 29: just state that there was excess NOx for VOC oxidation. As written the sentence is awkward.*

    **Response**: We revised the text to be the following:

    Lines 28-29: Under high-$NO_x$ conditions, NAC concentrations did not further increase with $NO_2$, while the $NO_3^-$ concentrations and ($NO_3^-$)/NACs ratios showed increasing trends…"

4.  *line 38: Where are the aerosol surface area data?*

    **Response**: The aerosol surface area data was described and analyzed in lines 141-143, 391-393 and Fig. 8(b).

    Lines 141-143: The aerosol surface area was calculated based on the measurements of particle number and size distribution by a scanning mobility particle sizer (SMPS, TSI 3936) and an aerosol particle sizer (APS, TSI 3321) (Yue et al., 2009; Wang et al., 2018a).

    Lines 391-393: The NAC concentrations also showed good correlations with aerosol surface area (Figure 8b). Higher aerosol surface area would facilitate the partitioning of gas-phase NAC

products or precursors into particle phase and the aqueous-phase or heterogeneous oxidation processes (Kroflic et al., 2015; Bauer et al., 2004; Fenter et al., 1996; Vidovic et al., 2018).

5. *lines 112-117: Please provide a chromatogram that shows the separation. Many of these compounds are isomers, and it is important to see evidence of sound chromatographic resolution. Evidence of peak quality for this study is warranted.*

**Responses**: The chromatograms of NAC standards and an ambient sample are added in Figure S1 (see general comments 2)). The NACs were identified and quantified based on the retention time of reference standards.

6. *Figure 1: these reaction schematics are not chemically balanced. If -$H_2O$ is removed than it should be indicated in the reaction scheme. What happens to the $NO_3$? This reaction scheme should show the reader exactly what happens chemically. Are these compounds also present in the gas-phase?*

**Response**: Only the major intermediate products related to NAC formation are shown in the schematic, and other intermediate products are omitted for clarity. The $NO_3$ radical would transform to $HNO_3$ in the reactions.

Yes. These compounds are present in both gas-phase and particle-phase, and both gas-phase and particle-phase pathways are included in Figure 1.

7. *line 162-165: This last sentence should be revised for clarity. Can't tell which conditions go with which season.*

**Response**: Revised accordingly (lines 209-211).

Lines 209-211: The NAC concentrations during summer (including this study) are generally lower than those during spring, autumn or winter, which could be due to stronger contributions from combustion sources (e.g. biomass burning and coal combustion) during spring, autumn or winter than those during summer.

8. *lines 173-174: So, it's a high NOx and a high anthropogenic VOC environment?*

**Response**: Yes. It's the environment characterized by both high-$NO_x$ and anthropogenic VOCs dominated conditions. This sentence has been revised to be clear (lines 223-225).

Lines 223-225: Thus, it is not a surprise to observe the high concentrations of nitrophenol and nitrocatechol in the typical high-$NO_x$ and anthropogenic VOCs dominated environments in summer in Beijing.

9. *lines 179-181: Please clarify the sentence about `Different NAC compositions'. How much higher is the RH in Hong Kong compared with Beijing in summer?*

**Response**: "Different NAC compositions" were described in lines 226-227: The contribution of NP among the total NACs at Changping was higher than that in summer in Hong Kong, while that of MNC was lower (Table S2 and the inserted pie charts in Figure S2).

The ambient RH in Hong Kong and Beijing were added in lines 232-234: The ambient RH in Hong Kong (>70%) was significantly higher than that in summer in Beijing (5-81%, 37% on average), thus the relative contribution of aqueous-phase pathways could be more dominant in Hong Kong, promoting the aqueous-phase formation of MNC.

10. *lines 205-209: It is unclear what is being discussed here with regard to the pathway, loss mechanisms, and groups. Please be specific and quantitative if possible.*

**Response**: The sentences have been revised (lines 265-270). The specific formation and loss pathways were analyzed in the following two paragraphs (lines 281-318).

Lines 265-270: Good inter-species correlations were observed among nitrophenol and its derivatives (2M4NP, 3M4NP, DMNP, r=0.56-0.88), as well as among nitrocatechol and its derivatives (3M6NC, 3M5NC, 4M5NC, r=0.49-0.84). This signaled that the formation and loss pathways as well as the influence factors were likely similar within NP and NC groups. In comparison, the correlations of NACs across the two groups, i.e., between nitrophenol derivatives (MNP, DMNP) and nitrocatechol derivatives (MNC, r=0.05-0.45), were lower (Table S3), suggesting different formation pathways and influence factors.

11. *lines 209-210: Why was the gas-phase not sampled for this study? It would have brought more clarity and we could have learned if the mass was simply distributed in the gas-phase as opposed to being 'lost'. It may be a good idea to use individual NAC concentrations, vapor pressures and day and night time temperature to estimate the total mass in both the gas- and particle-phase.*

**Response**: The gas-phase NACs were not measured due to the limitation of the samplers in this study. We have taken the reviewer's suggestion and estimated the gas-phase concentrations of NACs based on the equilibrium absorption partitioning theory. This new information indeed helps to explain the different behaviors of nitrophenols and nitrcatecols. The details have been provided in our response to the earlier comment (general comments 4)).

12. *line 271: please clarify `interacted conditions'*

**Response**: The interaction and reactions between anthropogenic VOCs and $NO_x$ are important pathways for the secondary formation of NACs in polluted urban atmospheres. This sentence was revised for clarity (lines 342-343).

Lines 342-343:…in polluted urban atmospheres that are characterized by high $NO_x$ and anthropogenic VOCs.

13. *lines 275-277: Please rewrite this sentence for clarity.*

**Response**: Revised accordingly (lines 345-347).

Lines 345-347: As only a limited number of VOC species were measured in this study, the $NO_x$ regime transition value was expressed by $NO_2$ concentrations rather than $NO_2$/VOC or $NO_x$/VOC ratios. We also note that the $NO_x$ regime transition values in other atmospheres could be quite different.

14. *Figure 6: While this figure is meant to indicate a general trend, it is unclear where the values in this figure came from. Are these averages? How much error are in these values? Please be specific.*

**Response**: Yes, these are the averages in each $NO_2$ concentration bin. The NAC compositions under similar $NO_2$ concentration levels were averaged, with a bin size of 10ppb $NO_2$ (lines 351-352). Figure 6 shows NAC compositions as a function of $NO_2$ levels. The standard deviations of the percentages (nitrocatechols among the total NACs or nitrophenols among the total NACs) were lower than 12% within each $NO_2$ bin (lines 353-355).

Lines 353-355: The contributions of NCs (standard deviation< 12% within each $NO_2$ bin) increased and those of NPs (standard deviation< 12% within each $NO_2$ bin) decreased at elevated $NO_2$ concentrations.

15. *lines 289-291: Again, a table containing the vapor pressures for these NAC compounds may help*

*explain some of these observations. Try the CHEMSPIDER web site.*

**Response**: The estimated vapor pressures of NACs are now added in Table S1.

*16. line 301: is this statistically `significant'?*

**Response**: Yes, this is statistically 'significant' at the 0.01 level. This is added in lines 370.

*17. 347-348: The last sentence in this paragraph is unclear.*

**Response**: Revised accordingly (lines 415-416)

Lines 415-416: The reaction mechanisms however are still unclear, which deserve further laboratory and field investigation in future studies.

---

## Author Comment (AC2) · 19 Apr 2019

Dear co-editor and referees,

We appreciate all your detailed and valuable comments on our manuscript of "acp-2018-1256". Please see the point-by-point response below and changes are marked blue in the revised manuscript.

**Referee #2**

**General comments**

*This work measured two groups of nitro-aromatic compounds (NACs), including nitrophenol (NP) and nitrocatechol (NC) like species at an urban site in summer Beijing. Their diurnal variations and formation pathways were discussed thoroughly with additional data of VOCs, meteorological parameters, etc. The authors found that the secondary formation is a more dominant NAC source than biomass burning, and the two types of NACs have distinct formation mechanisms. A concentration ratio of NACs was applied to indicate the relative importance of aqueous-phase and gas-phase oxidation. The NACs are an important group of light-absorbing compounds in ambient aerosols. Their compositions and sources in the atmosphere are very complex and studied intensively in recent years. The topic of this work is very interesting, and the manuscript is very well organized and written. However, some of the conclusions might need more work with more valid evidence.*

*1. Page 7, lines 194-198.*
*As NACs were more correlated with toluene and benzene (r = 0.64 – 0.70) than potassium (r = 0.49), the authors concluded that the NACs were more likely from photo-oxidation of VOCs than biomass burning in summer Beijing.*

*First, please provide the significance level of the correlations.*

**Response**: Significance levels of the correlations are added in lines 246-247 and Table S3.

Lines 246-247: It was noticed that NACs showed stronger correlations with toluene (r=0.70**), benzene (r=0.64**) or acetonitrile (r=0.61**) than those with potassium (r=0.49**).

*Second, the authors did regression analysis by comparing the particle-phase NACs with VOCs and potassium. In the atmosphere, NACs are mainly subject to two atmospheric processes, photolysis and gas-to-particle partitioning; toluene and benzene are in gas phase and precursors for SOA. The authors might need to consider the uncertainties in correlations due to atmospheric transformations.*

**Response**: The gas-phase NACs were calculated based on equilibrium absorption portioning theory (section 2.4 in lines 156-187, Table S1). The following sentence has been added to consider the uncertainties in correlations between particle-phase NACs and gas-phase precursors (lines 250-254).

Lines 250-254: We note that only particulate NAC concentrations were used to do the correlation analysis. Two atmospheric processes, namely photolysis and gas-to-particle partitioning, could influence the abundance of particle-phase NACs, especially for NP and MNPs, since majority of them was expected to be in the gas phase (Table S1). As such, correlations of particle-phase NP and MNPs with other species may less reliably reflect the underlying associations with the correlation species.

*2. Page 7, line 211 – 217*

*4M5NC and 3M5NC had higher nighttime concentrations than daytime, and were more correlated with RH than other NACs, which were ascribed to the aqueous-phase oxidation.*

*However, the authors might need to consider other possibilities.*

1). *Nitrocatechols are easier to be oxidized than nitrophenols. The second-order rate constant for nitrocatechol ($5 \times 10^9$ $M^{-1}$ $s^{-1}$) with OH in aqueous phase is much higher than nitrophenol ($3.7 \times 10^9$ $M^{-1}$ $s^{-1}$) (Hems and Abbatt, 2018). So the higher nighttime concentrations of 4M5NC and 3M5NC might also be associated with their shorter life time during daytime.*

**Response:** While we agree that the aqueous-phase OH oxidation of NCs is faster than NPs, this chemical loss pathway is likely negligible during daytime due to the low RH condition (average daytime RH=26%) and therefore low aerosol liquid water content.

2) *The nitrocatechol has one more hydroxyl group and is more water soluble than the nitrophenol. Besides gas-to-particle partitioning, the nitrocatechols might also be dissolved into particle water, which could be another explanation for the higher correlations between nitrocatechols and RH%.*

**Response**:   We have estimated the influence of NAC dissolution into particle water by Henry's law (lines 182-187). The contribution of NAC dissolution into aqueous-phase particles was estimated to be <0.02% to the total particulate NACs, indicating the dissolution pathway was negligible during our campaign. The following text is added to elaborate the contribution of the dissolution pathway.

Lines 182-187: Gas-phase NACs could also dissolve into the aqueous-phase particles. The concentrations dissolved into the aqueous phase ($C_{aq}$) were estimated by Henry's law (Sander, 2015). Henry constants were obtained from Sander et al. (2015) and ALWC was estimated using ISORROPIA-II (see section 2.4). The estimated $C_{aq}$ of 4NP and 3M4NP were 4.4E-4 and 2.4E-5 ng/m$^3$, contributing <0.02% to their concentrations in particle phase. The contribution of dissolution into aqueous-phase particles for NC and MNC is expected to be lower, due to the much lower gas-phase concentrations than that of 4NP. For this reason, we will not further consider the dissolution of NACs into particle aqueous phase.

3. *Page 8, lines 224- 226.*

*Could it be resulted from their different life times?*

**Response**: While we cannot exclude this possibility, we have not found evidence of their different life times. Previous study (Frka et al., 2016) suggested that the aqueous formation of 3M6NC need higher activation barriers than that of 3M5NC, which could be one reason for their different variations (lines 298-301).

4. *Page 9, line 250.*

*Why $NO_X$ oxidation of anthropogenic VOC precursors is the dominant source of NACs? Can we estimate the relative contributions of biomass burning and $NO_X$ oxidation?*

**Response**: Our analysis of inter-species correlations and temporal variations detailed in section 3.2 indicates that $NO_x$ oxidation of anthropogenic VOC precursors was the dominant source of NACs.

The relative contributions of $NO_x$ oxidation and primary biomass burning to NAC formation were not estimated in this study, since we do not have direct field measurement data for the differentiation. Addressing this question needs modeling studies incorporating emission inventories of the relevant VOC precursors. The following text (lines 246-259) is added to the manuscript to address this review comment.

Lines 246-259: It was noticed that NACs showed stronger correlations with toluene (r=0.70**), benzene (r=0.64**) or acetonitrile (r=0.61**) than those with potassium (r=0.49**). This

appeared to suggest that the $NO_x$ oxidation of anthropogenic VOCs and precursor VOCs from biomass burning was a more important source of NACs than primary biomass burning emission in summer in Beijing. A lower correlation between particulate NACs and EC (Table S3, r=0.39**) was also in agreement with the suggestion of the less importance of primary emissions to NACs during the campaign. …As for the relative importance of anthropogenic VOCs and biomass burning-derived VOCs, we do not have direct field measurement data for the differentiation. However, previous studies suggested that the sources of anthropogenic VOCs in summer in Beijing were dominated by vehicle emissions (>50%), with minor contributions from solvent evaporation and biomass burning (Wang et al., 2014; Liu et al., 2008). Modelling studies incorporating emission inventories of the relevant VOC precursors could address this issue and are suggested in future investigation of NAC sources.

*5. Page 10, lines 305-306. Page 11, lines 317-319.*

*As mentioned in comment 2, please consider the shorter life times of nitrocatechols than nitrophenols during the daytime, and the dissolution of nitrocatechols in particulate water as RH% increases.*

**Response**: Please see our earlier response to this issue (common 2).

*Moreover (connect to Page 6, lines 181-183), the authors thought that a higher relative humidity (RH%) would increase the relative contribution of aqueous-phase oxidation. Is the aqueous-phase oxidation in ambient aerosols only dominated by RH%?*

**Response**: The aqueous-phase oxidation in ambient aerosols is also influenced by aerosol acidity. It has been suggested that reaction pathways of nighttime formation of methyl-nitrocatechols via methylcatechol nitration are strongly pH dependent. The catechol oxidation-conjugated addition reaction mechanism can be an important pathway for dark secondary MNC formation in polluted environments with high NOx concentrations and relatively acidic particles (pH around 3) (Vidovic et al., 2018). Aerosol acidity was calculated using the ISORROPIA-II model in this study. During the campaign, particles were generally acidic with a pH range of 2.0-3.7, providing favourable conditions for the aqueous-phase oxidation formation of MNC (lines 283-289).

**Specific comments**

*1. Page 4, line 122. Please provide the recoveries for the target compounds.*

**Response**: The recoveries for target compounds were added in lines 129-130.

Lines 129-130: The recoveries of the target NACs were 91-106%.

**References**

Hems, R. F., and Abbatt, J. P. D.: Aqueous Phase Photo-oxidation of Brown Carbon Nitrophenols: Reaction Kinetics, Mechanism, and Evolution of Light Absorption, ACS Earth and Space Chemistry, 2, 225-234, 10.1021/acsearthspacechem.7b00123, 2018.

---

## Author Comment (AC3) · 19 Apr 2019

Dear co-editor and referees,

We appreciate all your detailed and valuable comments on our manuscript of "acp-2018-1256". Please see the point-by-point response below and changes are marked blue in the revised manuscript.

**Referee #3**

**General comments**

*Based on the extensive data from the field collected during the summertime campaign in Beijing, the manuscript discusses formation mechanisms of the most abundant groups of atmospheric nitroaromatic compounds, nitrophenols and nitrocatechols. The represented dataset is very valuable and the manuscript is well structured, however the conclusions made by the authors are sometimes vague and require more justification.*

*The authors distinguish between biomass-burning and anthropogenic emissions. As BB emissions are often of an anthropogenic origin, please make it clear that you mean traffic emissions when referring to anthropogenic emissions*

**Response**: In addition to vehicular exhaust, solvent evaporation could also be an anthropogenic precursor source for NACs. We have added the following text to clarify:

Lines 256-261: However, previous studies suggested that the sources of anthropogenic VOCs in summer in Beijing were dominated by vehicle emissions (>50%), with minor contributions from solvent evaporation and biomass burning (Wang et al., 2014; Liu et al., 2008). … We note that biomass burning could often be of an anthropogenic origin. Within this work, the term "anthropogenic VOCs" does not include VOCs from human-caused biomass burning activities.

**References**
Liu, Y., Shao, M., Fu, L., Lu, S., Zeng, L., and Tang, D.: Source profiles of volatile organic compounds (VOCs) measured in China: Part I, Atmos. Environ., 42, 6247-6260, 10.1016/j.atmosenv.2008.01.070, 2008.
Wang, M., Shao, M., Chen, W., Yuan, B., Lu, S., Zhang, Q., Zeng, L., and Wang, Q.: A temporally and spatially resolved validation of emission inventories by measurements of ambient volatile organic compounds in Beijing, China, Atmos. Chem. Phys., 14, 5871-5891, 10.5194/acp-14-5871-2014, 2014.

*In the introduction, much space is devoted to biomass burning as a possible source of NP and NC, however the characteristic methoxyphenols (substituted guaiacols and syringols) and their nitrated analogues are not even mentioned.*

**Response**: The NACs formation via the oxidation of methoxyphenols was added in the introduction (lines 62-65).

Lines 62-65: 4-Nitrocatechol could be formed via the OH-initiated oxidation of guaiacol, an abundant methoxyphenol emitted from biomass burning, in the presence of $NO_2$ (Lauraguais et al., 2014). However, under high-$NO_x$ conditions, this pathway seems to be of minor importance to nitrocatechol formation, instead, nitroguaiacols were formed as the major products (Lauraguais et al., 2014).

*Why don't you report the concentrations of (nitro)guaiacols and (nitro)syringols (maybe you didn't detect them?). This could help support the statement that anthropogenic (traffic?) emissions prevailed*

*over biomass burning emissions during the campaign. To my opinion, this assumption is not justified enough in the manuscript (although it is very likely to be true). Also, explain why primary emissions are not likely the important source of measured NAC.*

**Response**: (Nitro)guaiacols or (nitro)syringols were not detected in this study. Previous studies suggested that the sources of anthropogenic VOCs in summer in Beijing were dominated by vehicle emissions (>50%), with minor contributions from solvent evaporation and biomass burning (Wang et al., 2014; Liu et al., 2008). Modelling studies incorporating emission inventories of the relevant VOC precursors could address this issue and are suggested in future investigation of NAC sources (lines 256-259).

The explanation for why primary emissions are unlikely an important source of the measured NACs is provided in lines 246-250 and copied below for easy reference.

Lines 246-250: It was noticed that NACs showed stronger correlations with toluene (r=0.70**), benzene (r=0.64**) or acetonitrile (r=0.61**) than those with potassium (r=0.49**). This appeared to suggest that the $NO_x$ oxidation of anthropogenic VOCs and precursor VOCs from biomass burning was a more important source of NACs than primary biomass burning emission in summer in Beijing. A lower correlation between particulate NACs and EC (Table S3, r=0.39**) was also in agreement with the suggestion of the less importance of primary emissions to NACs during the campaign.

*In the cited paper of Frka et al. it is nicely demonstrated that the usage of surrogate molecules for the quantification of aromatic isomers by LC-MS can result in the over- or underestimation of the concentration of analyte, even for the factor of 10. The case of 3M4NC and 3M6NC is specifically exposed, as well as their misidentification in the past studies. Make it clear that you considered this issues during the analysis (if not, 3M6NC is more likely to be 3M4NC) and comment on the determined concentration of 3M6NC by use of 4M5NC standard. Reconsider also the conclusion in L224-226. Moreover, how do you know that DMNP peaks are truly dimethylnitrophenols and not ethylnitrophenols or their methoxylated isomers?*

**Response**: The over- or underestimation of 3M6NC and the possibility of misidentification 3M4NC as 3M6NC were described in lines 130-134. The conclusion in lines 224-226 was revised (lines 296-298). 2,6DM4NP was identified based on its retention time matching with that of authentic standards (Figure S1), while we cannot exclude the possibility of the other two DMNP peaks as ethylnitrophenols or methoxylated isomers (lines 135-137).

Lines 130-134: 4M5NC was employed as a surrogate standard to quantify 3M5NC and 3M6NC. However, a recent study suggested that no 3M6NC could be detected in ambient aerosols and the MNC isomer could be an incorrect assignment of 3M4NC as 3M6NC (Frka et al., 2016). We cannot exclude the possibility of MNC isomer as 3M4NC due to a lack of authentic standards. Employing 4M5NC as a surrogate standard, the concentrations of 3M6NC could be obviously underestimated due to its poor ionization under ESI condition compared with that of 4M5NC (Frka et al., 2016).

Lines 296-298: 3M6NC (or 3M4NC isomer) showed different temporal variations from 4M5NC or 3M5NC (Figures 4, S4) and their correlations were lower than that between 4M5NC and 3M5NC (Tables S3, S4), possibly suggesting different formation pathway for 3M6NC (or 3M4NC isomer) from those of 4M5NC or 3M5NC.

Lines 135-137: 2,6DM4NP was identified based on its retention time matching that of the authentic standard (Figure S1), while we cannot exclude the possibility of the other two DMNP isomers as ethylnitrophenols or methoxylated isomers.

*The language should be edited before the manuscript is published in ACP.*

**Response**: We have the manuscript edited and proofread by co-authors with extensive publishing experience in scientific journals.

**Specific comments**

*L60-62: how can MNC form by the oxidation of catechol? Did you mean methylcatechol?*

**Response**: Thanks for pointing out this mistake. Yes, we mean methylcatechols. The sentence is revised as below:

Line 60: Methyl-nitrocatechols (MNCs) could originate from $NO_x$ oxidation of methylated cresol or methylcatechols…

*L64-65: the sentence is misleading. In Fig. 1 you only show the most generally accepted pathways of gas-phase NAC formation. Revise the sentence for clarity and/or show also the aqueous-phase chemistry which you are referring to later in this paragraph. I strongly suggest extending the figure to at least the aqueous-phase formation of NC, which you show to be dominant in the atmosphere. Include also the relevant references in the figure caption.*

**Response**: Thanks for your suggestion. The figure has been extended to include the aqueous-phase formation of NCs and relevant references are included in the figure caption.

[Figure]

Figure 1 Formation pathways of nitrophenols, nitrocatechols, methyl-nitrophenols and methyl-nitrocatechols from the oxidation of benzene, toluene, phenol and methy-catechol in the gas phase and particle phase (Jenkin et al., 2003; Frka et al., 2016; Vione et al., 2004; Vione et al., 2001; Vidovic et al., 2018).

*L71-72: refer here to the very recent and comprehensive study on the possible nitration mechanisms of activated methoxyphenols in the atmospheric aqueous phase by Kroflič et al.: https://pubs.acs.org/doi/10.1021/acs.est.8b01903*

  **Response**: This study has been referred in line 74.

*L72: 'Nitrophenols could form through the nitration and hydroxylation of benzene…' note that NO₂ group deactivates the aromate, whereas OH makes it more reactive. Therefore, oxidation-first and nitration-second is more likely to occur. Change the order accordingly.*

  **Response**: Revised accordingly (lines 75-76).

  Lines 75-76: Nitrophenols could be formed through the hydroxylation and nitration of benzene in the presence of nitrite/nitrous acid or photo-nitration of phenol upon UV irradiation of nitrite in aqueous solutions.

*L93-94: I cannot see it obvious that anthropogenic emissions dominated during the campaign.*

**Response**: The obvious influence of anthropogenic pollutants was suggested in another study conducted during the campaign (Tang et al., 2018), which is referred here (lines 99). The sentence is re-phrased below to improve clarity:

  Lines 97-99: During this period, the site was influenced by anthropogenic pollutants from Beijing urban areas and under high-NO$_x$ conditions, as suggested by field measurement evidence reported in previous publications related to this campaign (Tang et al., 2018; Wang et al., 2018a).

*L149-150: check again, NP is also not always higher than in other studies.*

  **Response**: This sentence was revised (lines 195-197).

  Lines 195-197: Most NAC species (NC, MNP and MNC), except for DMNP and NP, also showed elevated concentrations in Changping, compared with those reported in other summertime studies (Figure S2).

*L192 (and Fig. 3 caption): what kind of pollution episodes?*

  **Response**: The pollution episodes were characterized by elevated organic concentrations. This sentence (lines 240-241) and Fig. 3 caption were revised for clarity.

  Lines 240-241: During the field campaign, four pollution episodes (episodes I, II, III, IV), marked by grey shading in Figure 3, were identified through observation of elevated organic aerosols.

  Fig. 3 caption: "…The pollution episodes, with elevated organic aerosols, are marked by gray shading."

*L210, L290, L326-327: It seems like photolysis is not important for the aqueous-phase products (NC)?*

  **Response**: Yes, we think the photolysis of aqueous-phase products (methyl-nitrocatechols) is of minor importance under low RH conditions during the campaign (average daytime RH=26%), compared with that of gas-phase products (nitrophenols). The gas-phase photolysis has been widely proved an important loss pathway of nitrophenols and methyl-nitrophenols (Yuan et al., 2016; Sangwan and Zhu, 2018; Bejan et al., 2007), with high fractions in gas phase. The aqueous-phase products (methyl-nitrocatechols) have been suggested to be dominated in particle phase. Under the low RH conditions (average daytime RH=26%) and low aerosols liquid water content (average daytime ALWC=2.9 μg/m$^3$) during daytime, the particles were at a solid to semisolid state (Liu et

al., 2017) and photochemical reactions in aqueous phase were largely suppressed. Thus, we think the photolysis of aqueous-phase products is of minor importance under our field campaign condition, which need more evidences to be validated in our future studies.

**References**

Bejan, I., Barnes, I., Olariu, R., Zhou, S., Wiesen, P., and Benter, T.: Investigations on the gas-phase photolysis and OH radical kinetics of methyl-2-nitrophenols, Phys. Chem. Chem. Phys., 9, 5686-5692, 10.1039/b709464g, 2007.

Liu, Y., Wu, Z., Wang, Y., Xiao, Y., Gu, F., Zheng, J., Tan, T., Shang, D., Wu, Y., Zeng, L., Hu, M., Bateman, A. P., and Martin, S. T.: Submicrometer Particles Are in the Liquid State during Heavy Haze Episodes in the Urban Atmosphere of Beijing, China, Environmental Science & Technology Letters, 4, 427-432, 10.1021/acs.estlett.7b00352, 2017.

Sangwan, M.and Zhu, L.: Role of Methyl-2-nitrophenol Photolysis as a Potential Source of OH Radicals in the Polluted Atmosphere: Implications from Laboratory Investigation, J. Phys. Chem. A, 122, 1861-1872, 10.1021/acs.jpca.7b11235, 2018.

Slade, J. H.and Knopf, D. A.: Multiphase OH oxidation kinetics of organic aerosol: The role of particle phase state and relative humidity, Geophys. Res. Lett., 41, 5297-5306, 10.1002/2014gl060582, 2014.

Yuan, B., Liggio, J., Wentzell, J., Li, S.-M., Stark, H., Roberts, J. M., Gilman, J., Lerner, B., Warneke, C., Li, R., Leithead, A., Osthoff, H. D., Wild, R., Brown, S. S., and de Gouw, J. A.: Secondary formation of nitrated phenols: insights from observations during the Uintah Basin Winter Ozone Study (UBWOS) 2014, Atmos. Chem. Phys., 16, 2139-2153, 10.5194/acp-16-2139-2016, 2016.

*L219-222: refer also to Fig. 9 – at the nightime RH levels, daytime ratios are also not dependent on RH anymore.*

**Response**: Yes. The daytime ratios are not dependent on RH anymore at the nighttime RH levels, which means the relative contribution of aqueous-phase and gas-phase pathways remain constant under higher RH conditions (the nighttime RH levels). While the concentrations of 3M4NC, 4M5NC and 4NP also increased at higher RH, the (3M4NC+ 4M5NC)/4NP ratio remained nearly constant. The aqueous-phase formation could be more dependent on ambient RH under lower RH levels during daytime.

*L259-260: 'NO$_3^-$ concentrations and (NO$_3^-$)/NACs ratios showed significant increasing trends (Figure 5a, b, c).' Not sure if I can see this, especially for the ratio. Please correct.*

**Response**: 'Significant' was deleted in the revised version. The NO$_3^-$ concentrations and (NO$_3^-$)/NACs ratios at NO$_2$ 20-25 ppb were higher than those under low-NOx conditions (NO$_2$<20 ppb) (lines 330-331).

Lines 330-331: At the same time, the NO$_3^-$ concentrations and (NO$_3^-$)/NACs ratios showed increasing trends compared with those under low-NO$_x$ conditions (NO$_2$< 20ppb) (Figure 5a, b, c).

*L325: Use here the original research of Frka et al. to avoid misunderstanding: https://pubs.acs.org/doi/abs/10.1021/acs.est.5b01811*

**Response**: The reference has been changed in the revised version (lines 393).

*Fig. 7: cannot find r value referred to in the caption.*

**Response**: This sentence was not supposed to be part of Figure 7 caption. We forget to delete this sentence in figure caption during the paper revision. This has been deleted in the revised version.

*Fig. S5: this is again only gas-phase chemistry presented. Make this crystal clear or add also heterogeneous/aqueous-phase pathways.*

**Response**: Figure S6 has been extended to include the aqueous-phase pathways.

[Figure]

Figure S6 Schematic diagram and simplified mechanisms of the competing formation of inorganic nitrates and NACs

**TECHNICAL CORRECTIONS**

*The usage of word 'obvious' is excessive. You should avoid it throughout the text.*

**Response**: The word 'obvious' were deleted where it's not clear or not necessary.

*It is not clear to me why you used Results and Discussion section 3 and later on again a Discussion section 4. I would use only section 3 – Results and Discussion.*

**Response**: Only section 3– Results and Discussion is now used in the revised version.

*L16 and L47: the phrase 'ultraviolet light absorption by brown carbon' is incorrect, the characteristics of BrC is visible light absorption. Please correct accordingly.*

**Response**: Revised accordingly (lines 46).

Lines 46: "…important contributors to the light absorption by brown carbon (BrC),…"

*L162-165: the sentence needs revision.*

**Response**: Revised accordingly (lines 209-211).

Lines 209-211: The NAC concentrations during summer (including this study) are generally lower than those during spring, autumn or winter, which could be due to stronger contributions from combustion sources (e.g. biomass burning and coal combustion) during spring, autumn or winter than those during summer.

*In general, the language, especially the grammar, requires editing.*

**Response**: Suggestion taken. We have edited the text to improve the language and grammar uses.

---

## Author Response (AR2)

Dear Dr. Willy Maenhaut,

Thank you very much for your technical suggestions on our manuscript of "acp-2018-1256". Please see the point-by-point responses below and changes are marked blue in the revised manuscript.

The authors have reasonably addressed the comments of the three anonymous referees and they have modified their manuscript accordingly. However, the comments given below should be addressed and several alterations are needed for the Main text and Supplement before the manuscript can be published in ACP.

For the Main text:

*Line 16: Replace ", however" by "; however".*
   **Response**: Changed accordingly.

*Line 18 and on 14 other occasions further within the text: "e.g. " by "e.g., ".*
   **Response**: Changed accordingly throughout the text.

*Lines 23-24: Replace "were found to be more dominant sources" by "was found to be a more dominant source".*
   **Response**: Changed accordingly.

*Line 31: Replace "indicating their different" by "indicating different".*
   **Response**: Changed accordingly.

*Line 33: Replace "pathways of" by "pathway of".*
   **Response**: Changed accordingly.

*Line 43: There should be a space after the 4 ";" in this line.*
   **Response**: Revised accordingly.

*Line 64: Replace ", instead" by "; instead".*
   **Response**: Changed accordingly.

*Line 69: Replace "and its derivatives" by "and their derivatives".*
   **Response**: Changed accordingly.

*Line 80: Replace "NACs is proposed" by "NACs are proposed".*
   **Response**: Changed accordingly.

*Line 85: Replace "of NAC," by "of NACs,".*
   **Response**: Changed accordingly.

*Line 109: It should be specified whether the aliquot was taken from the quartz fiber or the Teflon filter sample; I presume from the quartz fiber filter sample.*
   **Response**: Yes, the aliquot was taken from the quartz fiber filter sample. It has been added to be specific.

*Line 148: Replace "were analyzed" by "were measured". Furthermore, it should be specified*

*that the analyses for OC and EC were done on the quartz fiber filter samples.*
   **Response**: Revised accordingly.

*Line 150: It should be specified which filter samples were used for the IC analyses. I presume the Tefon filter samples. If not, what analyses were done on the Teflon filter samples? This should be indicated.*
   **Response**: Yes, teflon filter samples were used for the IC analyses. It has been added to be specific.

*Line 149: "Turpin and Lim, 2001" is missing in the Reference list.*
   **Response**: Added in the Reference list.

*Line 151: "Guo et al. (2010)" is missing in the Reference list.*
   **Response**: Added in the Reference list.

*Lines 158, 180 and 218: Replace "portitioning" by "partitioning".*
   **Response**: Changed accordingly.

*Lines 161-162: Replace "concentrations of organic aerosols (OA), calculated to be OC multiplying by" by "concentration of organic aerosols (OA), calculated as OC multiplied by".*
   **Response**: Changed accordingly.

*Line 166: Replace "saturated pressure" by "saturation pressure".*
   **Response**: Changed accordingly.

*Line 169: Some information (e.g., appropriate literature references) should be given for the Nannoolal approach and for the Joback and Reid approach.*
   **Response**: References have been added.

*Line 172: Replace "would not" by "will not".*
   **Response**: Changed accordingly.

*Line 175: "Cecinato et al. (2005)" is missing in the Reference list.*
   **Response**: Added in the Reference list.

*Lines 181 and 219: Replace "Le Breton et al. (2017)" by "Le Breton et al. (2018)".*
   **Response**: Changed accordingly.

*Lines 200-201: "Edwards et al., 2017" is missing in the Reference list.*
   **Response**: Added in the Reference list.

*Line 229: Replace "at two sites" by "at the two sites".*
   **Response**: Changed accordingly.

*Lines 246-247 and 249: It should be indicated what the "**" denote.*
   **Response**: The "**" denotes significant correlation at the 0.01 level, which has been indicated in the text.

*Line 247: Replace "than those with" by "than with".*
   **Response**: Changed accordingly.

*Line 264: Replace "didn't show" by "did not show".*

**Response**: Changed accordingly.

*Line 280: Replace "unlikely important" by "likely not important".*
   **Response**: Changed accordingly.

*Line 290: Replace "those during episodes" by "during episodes".*
   **Response**: Changed accordingly.

*Line 292: Replace "than the nighttime" by "than the nighttime ones".*
   **Response**: Changed accordingly.

*Line 298: Replace "The quantum" by "Quantum".*
   **Response**: Changed accordingly.

*Line 301: Replace "according to the" by "according to".*
   **Response**: Changed accordingly.

*Lines 303 and 305: Replace "et al (" by "et al. (".*
   **Response**: Changed accordingly.

*Line 307: Insert a space before "(Yuan".*
   **Response**: Revised accordingly.

*Line 309: Replace "play more" by "plays a more".*
   **Response**: Changed accordingly.

*Line 310: Replace "of campaign" by "of the campaign".*
   **Response**: Changed accordingly.

*Line 324: Replace "variation during" by "variations during".*
   **Response**: Changed accordingly.

*Line 329: Should it not be "anymore" instead of "any more" here?*
   **Response**: Revised accordingly.

*Line 338: Replace "higher anthropogenic" by "higher concentrations of anthropogenic".*
   **Response**: Changed accordingly.

*Line 342: Replace "existed in polluted" by "exist in polluted".*
   **Response**: Changed accordingly.

*Line 351: Replace "other, thus" by "other; thus".*
   **Response**: Changed accordingly.

*Line 352: Replace "10ppb" by "10 ppb".*
   **Response**: Changed accordingly.

*Line 358: Replace "represent the" by "represents the".*
   **Response**: Changed accordingly.

*Line 359: Replace "the increasing of" by "increasing".*
   **Response**: Changed accordingly.

*Line 368: Replace "values of toluene" by "value of toluene".*
   **Response**: Changed accordingly.

*Line 369: Replace "than those during" by "than during".*
   **Response**: Changed accordingly.

*Line 403: Replace "4-Nitrophenol" by "4-nitrophenol".*
   **Response**: Changed accordingly.

*Line 407: Replace "more important sources" by "a more important source".*
   **Response**: Changed accordingly.

*Line 414: Replace "atmospheres and that" by "atmospheres and indicates that".*
   **Response**: Changed accordingly.

*Line 415: Replace "mechanisms however are still unclear, which deserve" by "mechanisms are, however, still unclear and deserve".*
   **Response**: Changed accordingly.

*Line 418: Replace "4M5NC, daytime" by "4M5NC and daytime".*
   **Response**: Changed accordingly.

*Line 425: Replace "than those during" by "than during".*
   **Response**: Changed accordingly.

*Line 430: Replace "also important" by "also an important".*
   **Response**: Changed accordingly.

*Lines 452-637, References:*
*- Titles of journal articles should be in lower case and not in Title Case.*
*- For articles with 2 authors, the ", and" before the second author should be replaced by " and".*
   **Response**: References have been checked and revised accordingly.

*Line 534: Replace "Atmos. Chem. Phys., 1-32, 10.5194/acp-2017-814, 2017" by "Atmos. Chem. Phys., 18, 10355-10371, 10.5194/acp-18-10355-2018, 2018".*
   **Response**: Changed accordingly.

*Line 645: Replace "The summery" by "Summary" and replace "cross this" by "across this".*
   **Response**: Changed accordingly.

*Line 659: Replace "The concentrations" by "Concentration".*
   **Response**: Changed accordingly.

*Line 663: Replace "The variation of NAC compositions" by "Variation of NAC composition".*
   **Response**: Changed accordingly.

*Line 666: Replace "The concentrations" by "Concentration".*
   **Response**: Changed accordingly.

*Line 671: Replace "The correlation" by "Correlation".*
   **Response**: Changed accordingly.

*Line 675: Replace "Figure 9 The" by "Figure 9" replace "ratios as" by "ratio as".*
    **Response**: Changed accordingly.

For the Supplement:

*Page 2, line 1 of caption of Figure S2: Replace "The comparison" by "Comparison".*
    **Response**: Changed accordingly.

*Page 2, lines 2 and 6 of caption of Figure S2: Replace "Melptitz" by "Melpitz".*
    **Response**: Changed accordingly.

*Page 3, caption of Figure S3: Replace "The diurnal variations" by "Diurnal variation".*
    **Response**: Changed accordingly.

*Page 4, within Figure S5: Replace "dattime" by "daytime".*
    **Response**: Changed accordingly.

*Page 4, caption of Figure S6: Replace "The schematic diagram" by "Schematic diagram".*
    **Response**: Changed accordingly.

*Page 5, heading of Table S1: Replace "The saturated" by "Saturation".*
    **Response**: Changed accordingly.

*Page 6, heading of Table S2: Replace "The concentrations" by "Concentration".*
    **Response**: Changed accordingly.

*Page 6, within Table S2: Replace "Melptitz" by "Melpitz".*
    **Response**: Changed accordingly.

*Page 6, last line: Insert a space between "and" and "5-nitro-salicylic".*
    **Response**: Revised accordingly.

*Page 7, heading of Table S3: Replace "The pearson" by "Pearson".*
    **Response**: Changed accordingly.

*Page 7: There are data in bold, in red and in parentheses in red. It should be explained what this indicates.*
    **Response**: They have been explained following Table S3.

*Page 8, heading of Table S4: Replace "The pearson" by "Pearson".*
    **Response**: Changed accordingly.

*Pages 8 and 9: There are data bold, in red and in parentheses in red. It should be explained what this indicates.*
    **Response**: They have been explained following Table S4.

Most sincerely,

Min Hu and Jianzhen Yu

[revised manuscript text omitted]

Figure S2 Comparison of individual NAC species across this and previous studies. For each species, the concentrations are respectively reported in summer (yellow) in Melpitz, Germany (Teich et al., 2017), Flanders, Belgium (Kahnt et al., 2013), Xianghe, China (Teich et al., 2017), Wangdu, China (Teich et al., 2017), Ljubljana, Slovenia (Kitanovski et al., 2012), Hong Kong, China (Chow et al., 2016), Beijing, China (this study); in spring (green) in Flanders, Belgium (Kahnt et al., 2013), Hong Kong, China (Chow et al., 2016), Rome, Italy (Cecinato et al., 2005); in autumn (orange) in Flanders, Belgium (Kahnt et al., 2013), Hong Kong, China (Chow et al., 2016); and in winter (blue) in Melpitz, Germany (Teich et al., 2017), Detling, UK (Mohr et al., 2013), Flanders, Belgium (Kahnt et al., 2013), TROPOS, Germany (Teich et al., 2017) and Hong Kong, China (Chow et al., 2016) from left side to right side. The inserted pie charts represent the NAC compositions in summer in Hong Kong (Chow et al., 2016) and Changping, respectively.

[Figure]

Figure S3 Diurnal variation of (a) NO$_x$, (b) toluene, (c) benzene, (d) NO$_x$/BVOCs ratios and (e) NO$_x$/VOCs ratios.

[Figure]

Figure S4 Time series of (a) 4-nitrocatechol (4NC), (b) 4-methyl-6-nitrocatechol (4M6NC), and (c) 3-methyl-4-nitrophenol (3M4NP).

[Figure]

Figure S5 Day-night variations of NACs.

[Figure]

Figure S6 Schematic diagram and simplified mechanisms of the competing formation of inorganic nitrates and NACs

Table S1 Saturation pressure ($P_v$), NAC fraction in particles ($F_p$) and NAC concentrations in gas-phase

| compounds | $P_v$ [a](Pa) | $F_p$ (%) | | conc. in gas phase (ng/m$^3$) | |
|---|---|---|---|---|---|
| | | range | average | range | average |
| 4NP | 3.0E-02 | 0.25-1.3 | 0.83 | 143-566 | 268 |
| 2M4NP | 4.2E-03 | 1.2-7.6 | 5.1 | 5.2-42 | 15 |
| 3M4NP | 4.2E-03 | 1.2-7.6 | 5.1 | 1.9-20 | 5.8 |
| DMNP | 5.3E-04 | 99-100 | 100 | 0.000-0.007 | 0.002 |
| 4NC | 3.4E-05 | 67-91 | 85 | 0.06-0.79 | 0.27 |
| 3M6NC | 3.1E-06 | 95-99 | 98 | 0.000-0.003 | 0.001 |
| 3M5NC | 3.1E-06 | 95-99 | 99 | 0.002-0.017 | 0.007 |
| 4M5NC | 3.1E-06 | 95-99 | 98 | 0.001-0.028 | 0.009 |

Note: a. $P_v$ values were estimated by UManSysProp using approaches outlined in section 2.4.

Table S2 Concentration of nitroaromatic compounds in this and previous studies

[revised manuscript text omitted]

---

## Author Response (AR3)

Dear Dr. Willy Maenhaut,

Thank you again for your careful technical suggestions on our manuscript (acp-2018-1256). Please see the point-by-point responses below and changes are marked blue in the revised manuscript.

The following alterations are still needed for the Main text and Supplement before the manuscript can be published in ACP.

For the Main text:

*Line 148: Replace "on quartz fiber samples using" by "on the quartz fiber filter samples using a".*
*Line 150: Replace "Teflon filter" by "The Teflon filter".*
*Line 249: Replace "than that with" by "than with".*
*Line 311: Replace "play a more" by "plays a more".*
*Line 660: Replace "Summery of" by "Summary of".*
   **Response**: They have been changed accordingly in the Main text.

For the Supplement:

*Page 7, last line: Replace " negtive values" by "negative values".*
   **Response**: Changed accordingly in the Supplement.

Most sincerely,

Min Hu and Jianzhen Yu

[revised manuscript text omitted]

Figure S2 Comparison of individual NAC species across this and previous studies. For each species, the concentrations are respectively reported in summer (yellow) in Melpitz, Germany (Teich et al., 2017), Flanders, Belgium (Kahnt et al., 2013), Xianghe, China (Teich et al., 2017), Wangdu, China (Teich et al., 2017), Ljubljana, Slovenia (Kitanovski et al., 2012), Hong Kong, China (Chow et al., 2016), Beijing, China (this study); in spring (green) in Flanders, Belgium (Kahnt et al., 2013), Hong Kong, China (Chow et al., 2016), Rome, Italy (Cecinato et al., 2005); in autumn (orange) in Flanders, Belgium (Kahnt et al., 2013), Hong Kong, China (Chow et al., 2016); and in winter (blue) in Melpitz, Germany (Teich et al., 2017), Detling, UK (Mohr et al., 2013), Flanders, Belgium (Kahnt et al., 2013), TROPOS, Germany (Teich et al., 2017) and Hong Kong, China (Chow et al., 2016) from left side to right side. The inserted pie charts represent the NAC compositions in summer in Hong Kong (Chow et al., 2016) and Changping, respectively.

[Figure]

Figure S3 Diurnal variation of (a) NO$_x$, (b) toluene, (c) benzene, (d) NO$_x$/BVOCs ratios and (e) NO$_x$/VOCs ratios.

[Figure]

Figure S4 Time series of (a) 4-nitrocatechol (4NC), (b) 4-methyl-6-nitrocatechol (4M6NC), and (c) 3-methyl-4-nitrophenol (3M4NP).

[Figure]

Figure S5 Day-night variations of NACs.

[Figure]

Figure S6 Schematic diagram and simplified mechanisms of the competing formation of inorganic nitrates and NACs

Table S1 Saturation pressure ($P_v$), NAC fraction in particles ($F_p$) and NAC concentrations in gas-phase

| compounds | $P_v$ [a] (Pa) | $F_p$ (%) | | conc. in gas phase (ng/m$^3$) | |
|---|---|---|---|---|---|
| | | range | average | range | average |
| 4NP | 3.0E-02 | 0.25-1.3 | 0.83 | 143-566 | 268 |
| 2M4NP | 4.2E-03 | 1.2-7.6 | 5.1 | 5.2-42 | 15 |
| 3M4NP | 4.2E-03 | 1.2-7.6 | 5.1 | 1.9-20 | 5.8 |
| DMNP | 5.3E-04 | 99-100 | 100 | 0.000-0.007 | 0.002 |
| 4NC | 3.4E-05 | 67-91 | 85 | 0.06-0.79 | 0.27 |
| 3M6NC | 3.1E-06 | 95-99 | 98 | 0.000-0.003 | 0.001 |
| 3M5NC | 3.1E-06 | 95-99 | 99 | 0.002-0.017 | 0.007 |
| 4M5NC | 3.1E-06 | 95-99 | 98 | 0.001-0.028 | 0.009 |

Note: a. $P_v$ values were estimated by UManSysProp using approaches outlined in section 2.4.

Table S2 Concentration of nitroaromatic compounds in this and previous studies

[revised manuscript text omitted]

---

## Author Response (AR4)

Dear Dr. Willy Maenhaut,

Thank you again for your careful checking on our manuscript (acp-2018-1256). We are sorry for the spelling mistakes in the manuscript. Please see the point-by-point responses below and changes are marked blue in the revised manuscript.

The following alteration is still needed in the Supplement before the manuscript can be published in ACP:
*Page 9, last line: Replace " negtive values" by "negative values".*
   **Response**: Changed accordingly in the Supplement.

Most sincerely,

Min Hu and Jianzhen Yu

**Supplement for**

**The formation of nitro-aromatic compounds under high NO$_x$ and anthropogenic VOC conditions in urban Beijing, China**

Yujue Wang[1], Min Hu[*,1,5], Yuchen Wang[3], Jing Zheng[1], Dongjie Shang[1], Yudong Yang[1], Ying Liu[1,5], Xiao Li[1], Rongzhi Tang[1], Wenfei Zhu[6], Zhuofei Du[1], Yusheng Wu[1], Song Guo[1], Zhijun Wu[1], Shengrong Lou[6], Mattias Hallquist[2], and Jian Zhen Yu[*,3,4]

[1]State Key Joint Laboratory of Environmental Simulation and Pollution Control, College of Environmental Sciences and Engineering, Peking University, Beijing 100871, China

[2]Department of Chemistry and Molecular Biology, University of Gothenburg, Gothenburg, Sweden

[3]Environmental Science Programs, Hong Kong University of Science & Technology, Hong Kong, China

[4]Department of Chemistry, Hong Kong University of Science & Technology, Hong Kong, China

[5]Beijing Innovation Center for Engineering Sciences and Advanced Technology, Peking University, Beijing 100871, China

[6]Shanghai Academy of Environmental Sciences, Shanghai 200233, China

*Correspondence to*: Min Hu (minhu@pku.edu.cn); Jianzhen Yu (jian.yu@ust.hk)

[Figure]

Figure S1 Extracted ion chromatograms for (a) a standard solution containing 4-nitrocatechol (4NC), 4-nitrophenol (4NP), 2-methyl-4-nitrophenol (2M4NP), 3-methyl-4-nitrophenol (3M4NP), 2,6-dimethyl-4-nitrophenol (2,6DM4NP) and internal standard and (b) an ambient sample collected during the campaign.

[Figure]

Figure S2 Comparison of individual NAC species across this and previous studies. For each species, the concentrations are respectively reported in summer (yellow) in Melpitz, Germany (Teich et al., 2017), Flanders, Belgium (Kahnt et al., 2013), Xianghe, China (Teich et al., 2017), Wangdu, China (Teich et al., 2017), Ljubljana, Slovenia (Kitanovski et al., 2012), Hong Kong, China (Chow et al., 2016), Beijing, China (this study); in spring (green) in Flanders, Belgium (Kahnt et al., 2013), Hong Kong, China (Chow et al., 2016), Rome, Italy (Cecinato et al., 2005); in autumn (orange) in Flanders, Belgium (Kahnt et al., 2013), Hong Kong, China (Chow et al., 2016); and in winter (blue) in Melpitz, Germany (Teich et al., 2017), Detling, UK (Mohr et al., 2013), Flanders, Belgium (Kahnt et al., 2013), TROPOS, Germany (Teich et al., 2017) and Hong Kong, China (Chow et al., 2016) from left side to right side. The inserted pie charts represent the NAC compositions in summer in Hong Kong (Chow et al., 2016) and Changping, respectively.

[Figure]

Figure S3 Diurnal variation of (a) NO$_x$, (b) toluene, (c) benzene, (d) NO$_x$/BVOCs ratios and (e) NO$_x$/VOCs ratios.

[Figure]

Figure S4 Time series of (a) 4-nitrocatechol (4NC), (b) 4-methyl-6-nitrocatechol (4M6NC), and (c) 3-methyl-4-nitrophenol (3M4NP).

[Figure]

Figure S5 Day-night variations of NACs.

[Figure]

Figure S6 Schematic diagram and simplified mechanisms of the competing formation of inorganic nitrates and NACs

Table S1 Saturation pressure ($P_v$), NAC fraction in particles ($F_p$) and NAC concentrations in gas-phase

| compounds | $P_v$ [a](Pa) | $F_p$ (%) | | conc. in gas phase (ng/m$^3$) | |
|---|---|---|---|---|---|
| | | range | average | range | average |
| 4NP | 3.0E-02 | 0.25-1.3 | 0.83 | 143-566 | 268 |
| 2M4NP | 4.2E-03 | 1.2-7.6 | 5.1 | 5.2-42 | 15 |
| 3M4NP | 4.2E-03 | 1.2-7.6 | 5.1 | 1.9-20 | 5.8 |
| DMNP | 5.3E-04 | 99-100 | 100 | 0.000-0.007 | 0.002 |
| 4NC | 3.4E-05 | 67-91 | 85 | 0.06-0.79 | 0.27 |
| 3M6NC | 3.1E-06 | 95-99 | 98 | 0.000-0.003 | 0.001 |
| 3M5NC | 3.1E-06 | 95-99 | 99 | 0.002-0.017 | 0.007 |
| 4M5NC | 3.1E-06 | 95-99 | 98 | 0.001-0.028 | 0.009 |

Note: a. $P_v$ values were estimated by UManSysProp using approaches outlined in section 2.4.

Table S2 Concentration of nitroaromatic compounds in this and previous studies

[revised manuscript text omitted]